# Beyond Fine-Tuning: Transferring Behavior in Reinforcement Learning

## Abstract

Designing agents that acquire knowledge autonomously and use it to solve new tasks efficiently is an important challenge in reinforcement learning. Knowledge acquired during an unsupervised pre-training phase is often transferred by fine-tuning neural network weights once rewards are exposed, as is common practice in supervised domains. Given the nature of the reinforcement learning problem, we argue that standard fine-tuning strategies alone are not enough for efficient transfer in challenging domains. We introduce *Behavior Transfer* (BT), a technique that leverages pre-trained policies for exploration and that is complementary to transferring neural network weights. Our experiments show that, when combined with large-scale pre-training in the absence of rewards, existing intrinsic motivation objectives can lead to the emergence of complex behaviors. These pre-trained policies can then be leveraged by BT to discover better solutions than without pre-training, and combining BT with standard fine-tuning strategies results in additional benefits. The largest gains are generally observed in domains requiring structured exploration, including settings where the behavior of the pre-trained policies is misaligned with the downstream task.

## 1 Introduction

Transfer in deep learning is often performed through parameter initialization followed by fine-tuning, a technique that allows to leverage the power of deep networks in domains where labelled data is scarce [60, 16, 61, 22, 15]. This builds on the intuition that the pre-trained model will map inputs to a feature space where the downstream task is easy to perform. When combined with methods that can leverage massive amounts of unlabelled data for pre-training, this transfer strategy has led to unprecedented results in domains like computer vision [31, 30] and natural language processing [15, 50]. The success of these approaches has led to an ever-growing interest in developing techniques for pre-training large scale models on unlabelled data [9, 13, 24].

In the reinforcement learning (RL) context, unsupervised methods that learn in the absence of reward have also garnered much research attention [23, 21, 46, 19, 29]. The benefits of unsupervised pre-training are typically evaluated by their ability to enable efficient transfer to previously unseen reward functions [28]. In spite of their different approaches to unsupervised RL, most of the top-performing methods in this setting transfer knowledge through neural network weights. Such approaches deal with the data inefficiency associated to training neural networks with gradient descent, similarly to what is done in supervised learning, e.g. by pre-training encoders that extract representations from observations [59]. However, RL introduces a challenge that is not present in supervised learning: the agent is responsible for collecting the right data to learn from. This introduces a second source of inefficiency from which transfer approaches can also suffer if they rely on unstructured exploration strategies after pre-training, as these can lead to exponentially larger data requirements in complex

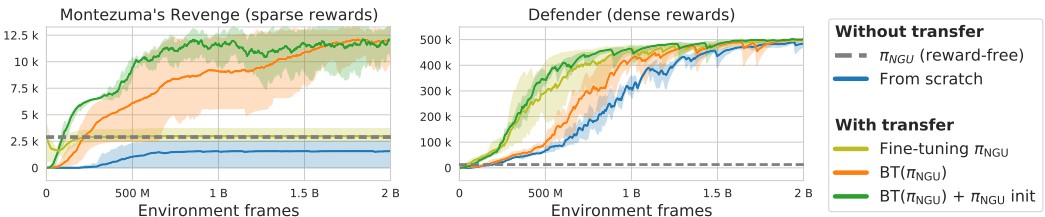

Figure 1: Comparison of transfer strategies on Montezuma's Revenge and Defender after pre-training a policy with NGU [48] in the absence of reward. The benefits of our proposed approach to leverage pre-trained behavior for exploration, Behavior Transfer (BT), are complementary to the gains provided by pre-trained weight initialization followed by fine-tuning.

downstream environments [45, 44]. To address this problem, one could consider fine-tuning policies that produce meaningful behavior [43, 52], but this approach quickly disregards the pre-trained behavior when learning in the downstream task due to catastrophic forgetting.

In this work, we explicitly separate the transfer of behaviour and weights. We propose to make use of the pre-trained behaviour itself (i.e., the pre-trained policy mapping from observations to actions) in contrast to pre-trained neural network weights for further fine-tuning. While pre-trained behavior has been used before for *exploitation* [5, 56, 2, 3], our approach employs pre-trained policies to aid with *exploration* as well to collect experience that can be leveraged via off-policy learning. This strategy accelerates learning, as the agent is exposed to potentially useful experience earlier in training, without compromising the quality of the discovered solution when the pre-trained behavior is not aligned with the downstream task. We expose the pre-trained behaviour to the downstream agent in two ways: firstly, as an extra exploratory strategy that, when randomly activated, persists for a number of steps, and secondly as an additional pseudo-action for the learned value function where the agent may elect to defer action selection to the pre-trained policy instead of choosing itself. We call this approach Behavior Transfer (BT).

Defining unsupervised RL objectives remains an open problem, and solutions are generally influenced by how the acquired knowledge will be used for solving downstream tasks. Instead of proposing yet another objective for unsupervised pre-training, we turn to existing techniques for training policies in the absence of reward and make our choice based on two general requirements. First, the objective should scale gracefully with increased compute and data. This has been key for the success of self-supervised approaches in other domains [9, 35], and we argue that it is an important property for unsupervised RL as well. Second, the pre-training stage should return a policy that produces complex behavior that may be leveraged in a subsequent transfer stage. The *Never Give Up* (NGU) [48] intrinsic reward meets both requirements, and our experiments show that large-scale pre-training with this objective leads to state of the art scores in the reward-free Atari benchmark.

Figure 1 exemplifies our main findings. We pre-train behaviour using the intrinsic NGU reward during a long unsupervised phase without rewards. This gives rise to exploratory behaviors that seek to visit many different states throughout an episode, and we then compare different strategies for leveraging the acquired knowledge once rewards are reinstated. While fine-tuning the pre-trained weights enables faster learning, the exploratory behavior of the pre-trained policy is quickly disregarded as it is exposed to rewards. On the other hand, Behavior Transfer (BT) does not modify the pre-trained policy while learning in the new task and is able to achieve higher end scores thanks to better exploration. These two strategies are not mutually exclusive, and BT also benefits from the faster convergence provided by initializing neural networks with pre-trained weights when these encode useful information for solving the downstream task.

Our contributions can be summarized as follows. (1) We propose *Behavior Transfer* (BT), a technique that leverages pre-trained policies for exploration by treating them as black boxes that are not modified during learning on the downstream task. BT uses the pre-trained policy to collect experience in two ways, namely randomly-triggered temporally-extended exploration and one-step calls based on value estimates. (2) Our experiments show that large-scale unsupervised pre-training with existing intrinsic rewards can produce meaningful behavior, achieving state of the art results in the reward-free Atari benchmark. These results suggest that scale is key for unsupervised RL, akin to what has been observed in supervised settings. (3) We provide extensive empirical evidence demonstrating the

benefits of leveraging pre-trained behavior via BT. Our approach obtains the largest gains in hard exploration games, where it almost doubles the median human normalized score achieved by our strongest baseline. Furthermore, we show that BT is able to leverage a single task-agnostic policy to solve multiple tasks in the same environment and to achieve high performance even when the pre-trained policies are misaligned with the task being solved. (4) BT brings benefits to the table that are complementary to those provided by reusing pre-trained neural network weights, and we empirically show that combining these two strategies can result in larger gains.

## 2  Preliminaries

The interaction between the agent and the environment is modelled as a Markov Decission Process (MDP) [49]. An MDP is defined by the tuple $(\mathcal{S}, \mathcal{A}, P, d_0, R, \gamma)$ where $\mathcal{S}$ and $\mathcal{A}$ are the state and action spaces, $P(s'|s, a)$ is the probability of transitioning from state $s$ to $s'$ after taking action $a$, $d_0(s)$ is the probability distribution over initial states, $R : \mathcal{S} \times \mathcal{A} \times \mathcal{S} \rightarrow \mathbb{R}$ is the reward function, and $\gamma \in [0, 1)$ is the discount factor. The goal is to find a policy $\pi(a|s)$ that maximizes the expected return, $G_t = \sum_{t=0}^{\infty} \gamma^t R_t$, where $R_t = r(S_t, A_t, S_{t+1})$. A principled way to address this problem is to use methods that compute action-value functions, $Q^\pi(s, a) = \mathbb{E}_\pi[G_t | S_t = s, A_t = a]$, where $\mathbb{E}_\pi[\cdot]$ denotes expectation over transitions induced by $\pi$ [49].

We consider a setting where the agent is allowed to first learn within an MDP without rewards, $\mathcal{M}^{\mathcal{R}} = (\mathcal{S}, \mathcal{A}, P, d_0)$, for a long period of time. The knowledge acquired during the reward-free stage is later leveraged when maximizing reward in new MDPs that share the same underlying dynamics but have different reward functions, $\mathcal{M}_i = (\mathcal{S}, \mathcal{A}, P, d_0, R_i, \gamma_i)$. Interactions between the agent and the environment are often assumed to incur a cost, but we will consider this cost to be relevant only for transitions with reward [28]. Even if the cost of unsupervised pre-training becomes non-negligible, it can be amortized when the acquired task-agnostic knowledge is leveraged to solve multiple tasks efficiently [15, 9]. Indeed, we would expect this transfer setting to become more relevant as the community moves towards more complex environments, where one may want to train agents to maximize multiple reward functions under constant dynamics. In the limit, one could consider the real world: it has constant or slowly changing dynamics, and humans are able to leverage previously acquired skills to quickly master new tasks.

## 3  Behavior Transfer

Transfer in supervised domains often exploits the fact that related tasks might be solved using similar representations. This practice deals with the data inefficiency of training large neural networks with stochastic gradient descent. However, there is an additional source of data inefficiency when training RL agents: unstructured exploration. Fine-tuning a pre-trained exploratory policy arises as a potential strategy for overcoming this problem, as the agent will observe rich experience much earlier in training than when initializing the policy randomly, but this approach suffers from important limitations. Learning in the downstream task can lead to catastrophically forgetting the pre-trained policy, thus prematurely disregarding its exploratory behavior. Moreover, the same neural network architecture needs to be used for both the pre-trained and the downstream policies, which in practice also imposes a limitation on the type of RL methods that can be employed in the adaptation stage (for instance, if the pre-trained policy was trained using a policy-based method, it might not be possible to fine-tune it using a value-based approach).

Let us assume that we have access to a pre-trained policy that exhibits exploratory behavior, and defer the discussion on how to train this policy to Section 4. Following such a policy might bring the agent to states that are unlikely to be visited with unstructured exploration techniques such as $\epsilon$-greedy [55]. This property has the potential of accelerating learning even when the behavior of the pre-trained policy is not aligned with the downstream task, as it will effectively shorten the path between otherwise distant states [41]. Leveraging pre-trained policies for exploration differs from other approaches in the literature that use such policies directly for exploitation, e.g. via zero-shot transfer [19], methods that define a higher-level policy that alternates between the given policies [5, 56], or within the framework of generalized policy updates [4]. Exploring with pre-trained policies can accelerate convergence by providing useful experience to the agent, which is possible even when the pre-training and downstream tasks are misaligned. However, strategies that directly use the pre-trained policies for exploitation may result in sub-optimal solutions in such scenario [2].

We propose to leverage the behavior of pre-trained policies during transfer to aid with exploration. An explicit distinction between behavior and representation is made by considering pre-trained policies as black boxes that take observations and return actions. This strategy is agnostic to how the pre-trained behavior is encoded and is not restricted to learned policies. We rely on off-policy learning methods during transfer to leverage the behavior of a pre-trained policy $\pi_p(a|s)$. We keep $\pi_p$ fixed during transfer, which prevents catastrophic forgetting of the original behavior when it is parameterized by a neural network (i.e., we instantiate and train a new policy with its own set of parameters). We propose *Behavior Transfer* (BT), which leverages two complementary strategies to achieve this. Since BT is agnostic to the method used to pre-train policies, $BT(\pi_p)$ refers to behavior being transferred from policy $\pi_p$. We formalize BT in the context of value-based Q-learning agents, although similar derivations are in principle possible for alternative off-policy learning methods. Pseudo-code for BT is provided in Algorithm 1.

**Temporally-extended exploration.** We draw inspiration from Lévy flights [57], a class of ecological models for animal foraging, where a fixed direction is followed for a duration sampled from a heavy-tailed distribution. This principle was implemented in the context of exploration in RL by $\epsilon z$-greedy [14], which encodes the notion of direction in the environment via exploration options that repeat the same action throughout the entire flight. Since $\pi_p$ is more likely to encode a meaningful notion of direction in complex environments than action repeats, we propose a variant of $\epsilon z$-greedy where $\pi_p$ is used as the exploration option. An exploratory flight might be started at any step with some probability. The duration for the flight is sampled from a heavy-tailed distribution (Zeta with $\mu = 2$ in all our experiments), and control is handed over to $\pi_p$ during the complete flight. When not in a flight, actions are sampled from the behavior policy obtained while maximizing the task reward (e.g. an $\epsilon$-greedy derived from the estimated Q values).

**Extra action.** The previous approach switches to $\pi_p$ during experience collection blindly, and we now consider an alternative strategy for triggering these switches based on value. This can be easily implemented through an extra action which samples an action from $\pi_p$, which also allows the agent to use the pre-trained policy at test time if deemed beneficial. More formally, this amounts to training a policy over an expanded action set $\mathcal{A}^+ = \mathcal{A} \cup \{a_+\}$, where $a_+$ is resolved by sampling an action from $\pi_p$, $a' \sim \pi_p(s)$ (with $a' \in \mathcal{A}$). The additional action can be seen as an option that can be initiated from any state and always terminates after a single step. Note that selecting the option will lead to the same outcome as if the agent had selected $a'$ as a primitive action, and we take advantage of this observation by using the return of following the option as target to fit both $Q(s, \pi_p(s))$ and $Q(s, a')$. Intuitively, this approach induces a bias that favours actions selected by $\pi_p$, accelerating the collection of rewarding transitions when the pre-trained policy is somewhat aligned with the downstream task. Otherwise, the agent can learn to ignore $\pi_p$ as training progresses by selecting other actions.

---

**Algorithm 1:** Experience collection pseudo-code for BT

**Input:** Action set, $\mathcal{A}$; additional action, $a_+$; extended action set, $\mathcal{A}^+ = \mathcal{A} \cup \{a_+\}$; pre-trained policy, $\pi_p$; Q-value estimate for the current policy, $Q^\pi(s, a) \, \forall a \in \mathcal{A}^+$; probability of taking an exploratory action, $\epsilon$; probability of starting a flight, $\epsilon_{\text{levy}}$; flight length distribution, $\mathcal{D}(\mathbb{N})$

**while** *True* **do**
    $n \leftarrow 0$                        // flight length
    **while** episode not ended **do**
        Observe state $s$
        **if** $n == 0$ and *random()* $\leq \epsilon_{levy}$ **then** $n \sim \mathcal{D}(\mathbb{N})$     // sample flight length
        **if** $n > 0$ **then**
            $n \leftarrow n - 1$
            $a \sim \pi_p(s)$
        **else**
            **if** *random()* $\leq \epsilon$ **then** $a \sim \text{Uniform}(\mathcal{A}^+)$ **else** $a \leftarrow \arg\max_{a' \in \mathcal{A}^+} [Q^\pi(s, a')]$
            **if** $a == a_+$ **then** $a \sim \pi_p(s)$
        **end**
        Take action $a$
    **end**
**end**

## 4 Reward-free pre-training

It is a common practice to derive objectives for proxy tasks in order to drive learning in the absence of reward functions, and there exists a plethora of different approaches in the literature. Model-based approaches can learn world models from unsupervised interaction [26]. However, the diversity of the training data will impact the accuracy of the model [53] and deploying this type of approach in visually complex domains like Atari remains an open problem [27]. Unsupervised RL has also been explored through the lens of *empowerment* [51, 42], which studies agents that aim to discover intrinsic options [23, 19]. While these options can be leveraged by hierarchical agents [21] or integrated within the universal successor features framework [2, 3, 8, 28], their potential lack of coverage generally limits their applicability to complex downstream tasks [12]. An alternative objective is that of exploring the environment by finding policies that induce maximally entropic state distributions [29, 39], although this might become extremely inefficient in high-dimensional state spaces without proper priors [40, 59].

Recall that our goal is to devise a pre-training objective that can help reduce the amount of interaction needed by the agent to collect relevant experience when learning in a downstream task. We argue that such objective needs to meet two requirements. First, as suggested by results in other domains [9, 35], it should scale gracefully as the amount of compute and experience used for pre-training are increased. This contrasts with the training regimes used in most unsupervised RL approaches, which use a relatively small amount of experience [28, 40, 59] when compared to distributed agents that do make use of rewards [33, 18, 36]. Second, it must encourage the emergence of complex behaviors such as navigation or manipulation skills. It has been argued that exploring the environment efficiently will serve as a proxy for developing such behaviors [37], and exploration bonuses have been shown to produce meaningful behavior in the absence of reward [46, 10]. However, many exploration bonuses vanish over the course of training and thus may not be well-suited for a long unsupervised pre-training phase. It can be shown that many intrinsic rewards aim at maximizing the entropy of all states visited during training, and so the final policy does not necessarily exhibit exploratory behavior [39].

We propose to use Never Give Up (NGU) [48] as a means for training exploratory policies in an unsupervised setting. The NGU intrinsic reward proposes a curiosity-driven approach for training persistent exploratory policies which combines per-episode and life-long novelty. The per-episode novelty, $r_t^{\text{episodic}}$, rapidly vanishes over the course of an episode, and it is designed to encourage self-avoiding trajectories. It is computed by comparing a representation of the current observation, $f(s_t)$, to those of all the observations visited in the current episode, $M = \{f(s_0), f(s_1), \ldots, f(s_{t-1})\}$, where $f : \mathcal{S} \to \mathbb{R}^p$ is an embedding function trained using a self-supervised inverse dynamics model [46]. Such a mapping concentrates on the controllable aspects of the environment, ignoring all the variability present in the observation that is not affected by the action taken by the agent. The life-long novelty, $\alpha_t$, slowly vanishes throughout training, and it is computed by using Random Network Distillation (RND) [11]. With this, the intrinsic reward $r_t^{\text{NGU}}$ is defined as follows:

$$r_t^{\text{NGU}} = r_t^{\text{episodic}} \cdot \min\left\{\max\left\{\alpha_t, 1\right\}, L\right\}, \ \text{with} \ \ r_t^{\text{episodic}} = \frac{1}{\sqrt{\sum_{f(s_i) \in N_k} K(f(s_t), f(s_i))} + c} \quad (1)$$

where $L$ is a fixed maximum reward scaling, $N_k$ is the set containing the $k$-nearest neighbors of $f(s_t)$ in $M$, $c$ is a constant and $K : \mathbb{R}^p \times \mathbb{R}^p \to \mathbb{R}^+$ is a kernel function satisfying $K(x, x) = 1$ (which can be thought of as approximating pseudo-counts [48]). The episodic component of the reward in Equation 1 is reset by emptying $M$ with each episode, thus the NGU reward does not vanish throughout the training process. This makes it suitable for driving learning in task-agnostic settings. Further details on NGU are reported in the supplementary material.

## 5 Experiments

Agents are evaluated in the Atari suite [7], a benchmark that presents a variety of challenges and that is a common test ground for RL agents with unsupervised pre-training [28, 40, 52]. Experiments are run using the distributed R2D2 agent [36] with 256 CPU actors and a single GPU learner. Policies use the same Q-Network architecture as Agent57 [47], which is composed by a convolutional torso followed by an LSTM [32] and a dueling head [58]. Hyperparameters and a detailed description of the full distributed setting are provided in the supplementary material. All reported results are the average over three random seeds.

**Reward-free learning.** The amount of task reward collected by unsupervised policies is often used as a proxy to measure their quality [19]. While the actual utility of these policies will not be revealed until they are leveraged for transfer, this proxy lets us evaluate whether the discovered behavior changes as longer pre-training budgets are allowed. We compare unsupervised NGU policies against VISR [28] and APT [40], which utilize a small amount of supervised interaction to adapt the pre-trained policies. We also consider two additional unsupervised baselines: *(i)* a constant positive reward at each timestep that favours long episodes, which correlate with high scores in some games [10], and *(ii)* RND [11], which rewards life-long novelty. Note that the RND reward vanishes, but we include it in our analysis because it was previously used by Burda et al. [10] in this setting and implementation choices such as reward normalization may prevent it from fading in practice. Figure 2 (left) shows how the zero-shot transfer performance of unsupervised policies evolves during a long pre-training phase. NGU reaches the highest scores, but both NGU and RND eventually outperform VISR and APT even though these used supervised interaction. In Table 2 of Appendix C we show that unsupervised NGU policies largely outperform several other baselines using the standard pre-training and adaptation setting. These results highlight the importance of large-scale unsupervised pre-training in RL, similarly to the trend observed in supervised domains [9].

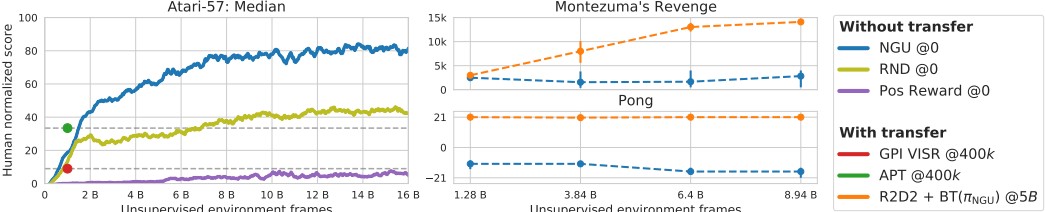

Figure 2: Performance as a function of the pre-training budget. $@N$ represents the number of frames with reward utilized for transfer. **(Left)** Median human normalized score across the 57 games in the Atari suite. We observe the emergence of useful behavior when optimizing an intrinsic reward during a long unsupervised pre-training of 16B frames, which contrasts with the shorter pre-training of 1B frames in previous works [28, 40]. **(Right)** Scores in the games of Montezuma's Revenge (sparse rewards) and Pong (dense reward), before and after transfer, as a function of the pre-training budget. A longer pre-training benefits transfer in hard exploration games even if the zero-shot transfer score of the unsupervised policies does not increase.

**Transfer setting.** Transfer approaches are typically evaluated in the Atari benchmark with a budget of 100k RL interactions with reward (400k frames), but we propose to allow a longer adaptation phase. Randomly initialized networks tend to overfit in these very low data regimes without strong regularization [38], and we are interested in studying the impact of leveraging behavior both in isolation and combined with transfer via pre-trained weights. Moreover, since the pre-trained policies are already competent in the downstream tasks, 100k interactions are exhausted after few episodes and may be insufficient for improving performance. For these reasons, we provide results with up to 1.25B RL steps of supervised interaction (5B frames). This allows evaluating both convergence speed and asymptotic performance, while still being a relatively small budget for these distributed agents with hundreds of actors [47].

**Transfer via behavior.** We start by studying the impact of leveraging behavior in isolation, i.e. without transferring pre-trained weights, when learning in downstream tasks. We compare BT against two baselines that do not use pre-trained behavior, namely the standard R2D2 agent [36] that uses $\epsilon$-greedy policies for exploration [55], as well as a variant of R2D2 with $\epsilon z$-greedy exploration [14]. Figure 3 shows that BT is superior to both baselines for any amount of environment interaction with rewards, converging faster early in training and also obtaining higher asymptotic performance. These results also demonstrate the generality of the proposed approach, as it is able to benefit from both RND and NGU policies. Note that BT performs particularly well in the set of six hard exploration games[1] defined by Bellemare et al. [6], which is aligned with our intuition that reusing behavior helps overcoming the inefficiency associated to unstructured exploration. Figure 2 (right) confirms that a long pre-training phase is especially important in hard exploration games such as Montezuma's Revenge, even it they do not translate into higher zero-shot transfer scores, as it produces more exploratory behavior. On the other hand, the performance after transfer is independent of the amount of pre-training in dense reward games like Pong, where unstructured exploration is enough to reach optimal scores.

---

[1]`gravitar`, `montezuma_revenge`, `pitfall`, `private_eye`, `solaris`, `venture`

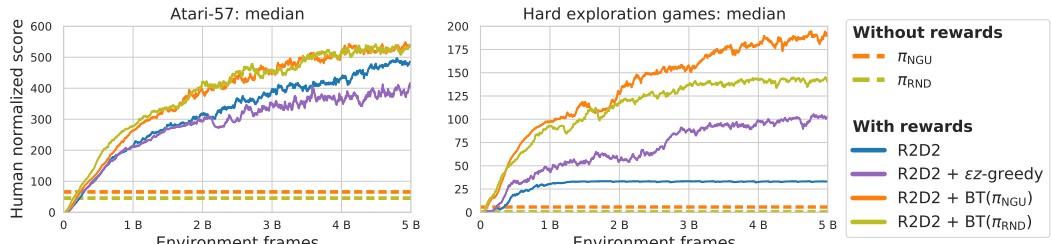

Figure 3: Median human normalized scores for R2D2-based agents trained from scratch. (**Left**) Full Atari suite. (**Right**) Subset of hard exploration games.

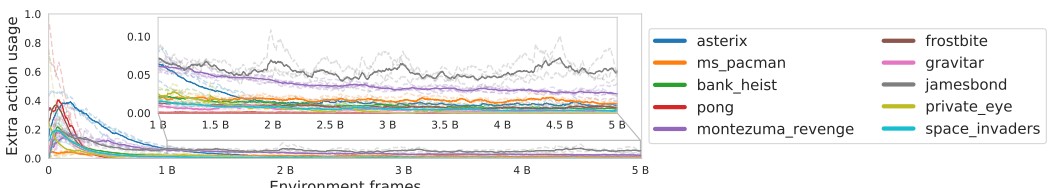

Figure 4: Usage of the extra action in $\text{BT}(\pi_{\text{NGU}})$, computed as the fraction of steps within an episode in which it is selected by the agent. The usage peaks early in training and slowly decreases afterwards as the new policy becomes stronger at the task.

**Ablation studies.** In order to gain insight on each of the components in BT, we run experiments on a subset of 12 games[2] requiring different amounts of exploration and featuring both dense and sparse rewards. $\text{BT}(\pi_{\text{NGU}})$ achieves a median score of 368 in this subset, which compares favorably to the 196 median score of R2D2 with $\epsilon$-greedy exploration. Removing either the extra action or the temporally-extended exploration reduces the median score of $\text{BT}(\pi_{\text{NGU}})$ to 224. These results suggest that the gains provided by both strategies are complementary, and both are responsible for the strong performance of BT. To provide further insight about the benefits of BT, Figure 4 reports the fraction of steps per episode in which the extra action is selected by the greedy policy. It hints at the emergence of a schedule over the usage of the pre-trained policy, which increases early in training and decays afterwards. We hypothesize that this is due to the fact that the unsupervised policies obtain large episodic returns, but their behavior is suboptimal when maximizing discounted rewards. These policies take many exploratory actions in between rewards, and so the agent eventually figures out more efficient strategies for reaching rewarding states by using primitive actions.

**Transfer to multiple tasks.** An appealing property of task-agnostic knowledge is that it can be leveraged to solve multiple tasks. In the RL setting, this can be evaluated by leveraging a single task-agnostic policy for solving multiple tasks (i.e. reward functions) in the same environment. We evaluate whether the unsupervised NGU policies can be useful beyond the standard Atari tasks by creating two alternative versions of Ms Pacman and Hero with different levels of difficulty. The goal in the modified version of Ms Pacman is to eat vulnerable ghosts, with pac-dots giving $0$ (easy version) or $-10$ (hard version) points. In the modified version of Hero, saving miners gives a fixed return of 1000 points and dynamiting walls gives either $0$ (easy version) or $-300$ (hard version) points. The rest of rewards are removed, e.g. eating fruit in Ms Pacman or the bonus for unused power units in Hero. Note that even in the easy version of the games exploration is harder than in their original counterparts, as there are no small rewards guiding the agent towards its goals. Exploration is even more challenging in the hard version of the games, as the intermediate rewards work as a deceptive signal that takes the agent away from its actual goal. In this case, finding rewarding behaviors requires a stronger commitment to an exploration strategy. Unsupervised NGU policies often achieve very low or even negative rewards in this setting, which contrasts with the strong performance they showed when evaluated under the standard game reward. Figure 5 shows that leveraging the behavior of pre-trained exploration policies provides important gains even in this adversarial scenario. These results suggest that the strong performance observed under the standard game rewards is not due to an

---

[2]Obtained by combining games used to tune hyperparameters in [28] with games where $\epsilon z$-greedy provides clear gains over $\epsilon$-greedy as per [14]: `asterix`, `bank_heist`, `frostbite`, `gravitar`, `jamesbond`, `montezuma_revenge`, `ms_pacman`, `pong`, `private_eye`, `space_invaders`, `tennis`, `up_n_down`.

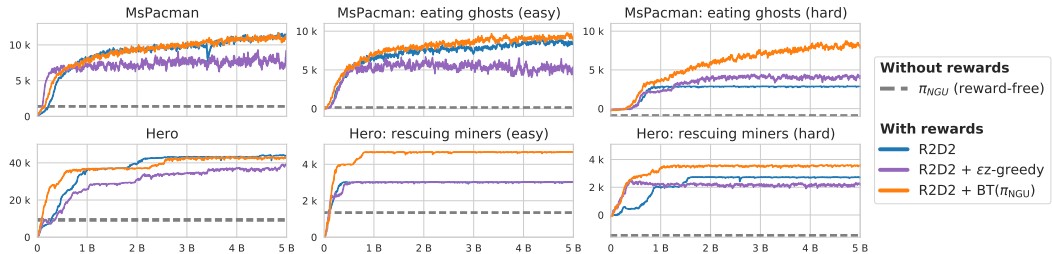

Figure 5: Scores in Atari games with modified reward functions. We train a single task-agnostic policy per environment, and leverage it to solve three different tasks: the standard game reward, a task with sparse rewards (easy), and a variant of the same task with deceptive rewards (hard).

alignment between the NGU reward and the game goals, but due to an efficient usage of pre-trained exploration policies.

**Combining pre-trained behavior and weights.** Our last batch of experiments focuses on studying transfer via pre-trained weights and its compatibility with BT. Policies are composed of a convolutional torso, an LSTM, and a dueling head. We consider two initialization strategies: a *partial initialization* approach that loads the torso and the LSTM, but initializes the head randomly; and a *full initialization* scheme where all weights are loaded. The former can be understood as transferring learned representations [59], but deferring exploration to a random policy. On the other hand, the full initialization approach can be seen as directly transferring the policy and is usually referred to as fine-tuning the pre-trained policy [43, 40, 52]. Note that these approaches only change how weights are initialized *before* training. As in previous experiments, all parameters in the new policy are trained and $\pi_p$ is kept fixed when using BT. Figure 6 (top) compares agents with and without BT for different amounts of transfer via weights on the Atari benchmark. Loading pre-trained weights results in faster learning early in training, both with and without BT. The largest gains are observed in dense reward games, which translates into higher median scores across the full suite because most games belong to this category. Weights alone are not enough in hard exploration games, where leveraging the pre-trained policy via BT provides clear benefits. Perhaps surprisingly, we observe that transferring representations outperforms fine-tuning the pre-trained policy, and we hypothesize that the former is more robust to misalignments between the pre-trained policy and the downstream task. This intuition is further supported by the experiments on games with modified reward functions reported in Figure 6 (middle & bottom), where the faster learning provided by pre-trained weights often comes at the cost of lower end scores. On the other hand, BT is crucial in tasks with sparse and deceptive rewards and also benefits from pre-trained weights in tasks where positive transfer is observed.

# 6 Related work

Our work uses the experimental methodology presented by Hansen et al. [28]. Whereas that work only considered a fast, simplified adaptation process that limited the final performance on the downstream task, we focus on the more general case of using a previously trained policy to aid in solving the full RL problem. Hansen et al. [28] use successor features to identify which of the pre-trained tasks best matches the true reward structure, which has previously been shown to work well for multi-task transfer [3]. Bagot et al. [1] augments an agent with the ability to utilize another policy, which is learned in tandem based on an intrinsic reward function. This promising direction is complementary to our work, as it handles the case wherein there is no unsupervised pre-training phase.

Gupta et al. [25] provides an alternative method to meta-learn a solver for reinforcement learning problems from unsupervised reward functions. This method utilizes gradient-based meta-learning [20], which makes the adaptation process standard reinforcement learning updates. This means that even if the downstream reward is far outside of the training distribution, final performance would not necessarily be affected. However, these methods are hard to scale to the larger networks considered here, and followup work [34] changed to memory-based meta-learning [17] which relies on information about rewards staying in the recurrent state. This makes it unsuitable to the sort of hard exploration problem our method excels at. Recent work has shown success in transferring representations learned in an unsupervised setting to reinforcement learning tasks [54]. Our representation transfer experi-

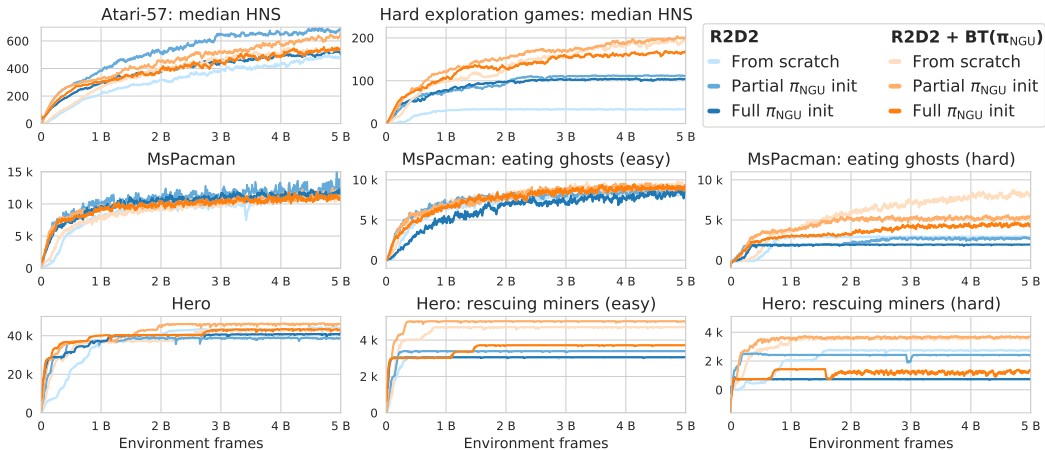

Figure 6: Performance of R2D2-based agents with different amounts of transfer via weights. Policies are composed of a CNN encoder followed by an LSTM and a dueling head. We compare training from scratch, loading all weights (Full $\pi_{\text{NGU}}$ init) or all weights except those in the dueling head (Partial $\pi_{\text{NGU}}$ init). (**Top**) Median human normalized scores (HNS) in the full Atari suite (left) and the subset of hard exploration games (right). (**Middle & Bottom**) Games with modified reward functions as in Figure 5.

ments suggest that this might handicap final performance, but the possibility also exists that different unsupervised objectives should be used for representation transfer and policy transfer.

## 7    Discussion

We studied the problem of transferring pre-trained behavior for exploration in reinforcement learning, an approach that is complementary to the common practice of transferring neural network weights. Our proposed approach, Behavior Transfer (BT), relies on the pre-trained policy for collecting experience in two different ways: (*i*) through temporally-extended exploration, which can be triggered with some probability at any step, and (*ii*) via one-step calls to the pre-trained policy based on value estimates. BT results in strong transfer performance when combined with exploratory policies pre-trained in the absence of reward, with the most important gains being observed in hard exploration tasks. These benefits are not due to an alignment between our pre-training and downstream tasks, as we also observed positive transfer in games where the pre-trained policy obtained low scores. In order to provide further evidence for this claim, we designed alternative tasks for Atari games involving hard exploration and deceptive rewards. Our transfer strategy outperformed all considered baselines in these settings, even when the pre-trained policy obtained very low or even negative scores, demonstrating the generality of the method. Besides disambiguating the role of the alignment between pre-training and downstream tasks, these experiments demonstrate the utility of a single task-agnostic policy for solving multiple tasks in the same environment. Finally, we also demonstrated that BT can be combined with transfer via neural network weights to provide further gains.

Our experimental results highlight the importance of scale when training RL agents in reward-free settings, which is one of the key factors behind the recent success of unsupervised approaches in other domains. This contrasts with the small budgets considered for reward-free RL in previous works and motivates further research in unsupervised RL approaches that scale with increased data and compute. We argue that scale is one of the missing components in reward-free RL, and it will be a necessary condition to unfold its full potential. Beyond improving the unsupervised learning phase, we are also excited about the possibilities unlocked by BT and that are not possible when transferring knowledge through weights, such as leveraging multiple pre-trained policies and deploying BT in continual learning scenarios where the agent never stops learning and keeps accumulating knowledge and skills. Future work should also study improved mechanisms for handing over control to pre-trained policies, as well as prioritizing the usage of certain behaviors over others when multiple such policies are available to the agent. This could overcome one of the current limitations of BT, which assumes that flights can be started from any state and still produce meaningful behavior.

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
