# A  Pseudo-code

Algorithm 2 provides pseudo-code for the flight logic that controls how the pre-trained policy is used for temporally-extended exploration. At each step, a flight is started with probability $\epsilon_{\text{levy}}$. The duration of the flight is sampled from a heavy-tailed distribution, $\mathcal{D}(\mathbb{N})$, similarly to $\epsilon z$-greedy (c.f. Appendix B for more details). When not in a flight, the exploitative policy that maximizes the extrinsic reward is derived from the estimated Q-values using the $\epsilon$-greedy operator. This ensures that all state-action pairs will be visited given enough time, as exploring only with $\pi_p$ does not guarantee such property.

Algorithm 3 provides pseudo-code for the actor logic when using the augmented action set, $\mathcal{A}^+ = \mathcal{A} \cup \{\pi_p(s)\}$. It derives an $\epsilon$-greedy policy over $|\mathcal{A}| + 1$ actions, where the $(|\mathcal{A}| + 1)$-th action is resolved by sampling from $\pi_p(s)$.

**Algorithm 2:** Experience collection pseudo-code for BT with temporally-extended exploration

**Input:** Action set $\mathcal{A}$
**Input:** Q-value estimate for the current policy, $Q^\pi(s, a) \, \forall a \in \mathcal{A}$
**Input:** Pre-trained policy, $\pi_p$
**Input:** Probability of starting a flight, $\epsilon_{\text{levy}}$
**Input:** Flight length distribution, $\mathcal{D}(\mathbb{N})$
**while** *True* **do**

    $n \leftarrow 0$                         `// flight length`

    **while** episode not ended **do**

        Observe state $s$

        **if** $n == 0$ and *random()* $\leq \epsilon_{levy}$ **then**

            $n \sim \mathcal{D}(\mathbb{N})$          `// sample from distribution over lengths`

        **end**

        **if** $n > 0$ **then**

            $n \leftarrow n - 1$

            $a \sim \pi_p(s)$

        **else**

            $a \sim \epsilon\text{-greedy}[Q^\pi(s, a)]$

        **end**

        Take action $a$

    **end**

**end**

---

**Algorithm 3:** Experience collection pseudo-code for BT with an extra action

**Input:** Action set $\mathcal{A}$
**Input:** Additional action, $a_+$
**Input:** Extended action set, $\mathcal{A}^+ = \mathcal{A} \cup \{a_+\}$
**Input:** Pre-trained policy, $\pi_p$
**Input:** Q-value estimate for the current policy, $Q^\pi(s, a) \, \forall a \in \mathcal{A}^+$
**Input:** Probability of taking an exploratory action, $\epsilon$
**while** *True* **do**

    **while** episode not ended **do**

        Observe state $s$

        **if** *random()* $\leq \epsilon$ **then**

            $a \sim \text{Uniform}(\mathcal{A}^+)$

        **else**

            $a \leftarrow \arg\max_{a' \in \mathcal{A}^+}[Q^\pi(s, a')]$

        **end**

        **if** $a == a_+$ **then**

            $a \sim \pi_p(s)$

        **end**

        Take action $a$

    **end**

**end**

## B Hyperparameters

All policies use the same Q-Network architecture as Agent57 [51], which is composed by a convolutional torso followed by an LSTM [32] and a dueling head [62]. When leveraging the behavior of the pre-trained policy to solve new tasks, we instantiate a new network with independent weights (c.f. Figure 7). One can initialize some of the components of the new network using pre-trained weights without tying their values (as in common fine-tuning approaches).

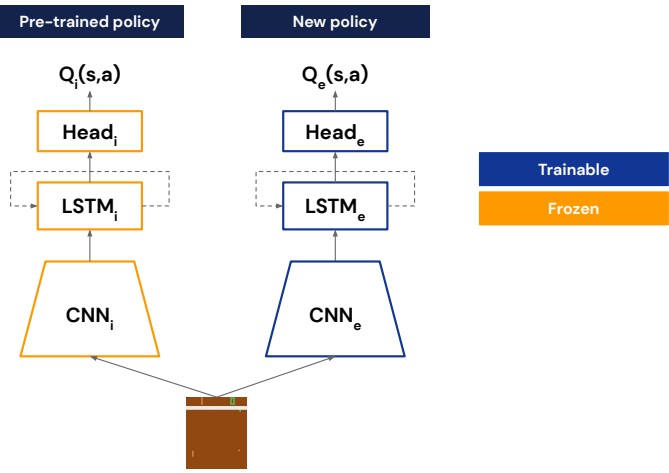

Figure 7: Q-Network architecture for the reinforcement learning stage. The networks use independent sets of parameters, and the weights of the pre-trained policy are kept fixed to preserve the learned behavior.

Table 1 summarizes the main hyperparameters of our method. The pre-trained policies were optimized using Retrace [45]. Learning with rewards was performed with Peng's $Q(\lambda)$ [50] instead, which we found to be much more data efficient in our experiments. The reason for this difference is that the benefits of $Q(\lambda)$ were observed once unsupervised policies had been trained on all Atari games.

Table 1: Hyperparameter values used in R2D2-based agents. The rest of hyperparameters use the values reported by Kapturowski et al. [37].

| Hyperparameter | Value |
| --- | --- |
| Number of actors | 256 |
| Actor parameter update interval | 400 environment steps |
| Sequence length | 160 (without burn-in) |
| Replay buffer size | $12.5 \times 10^4$ part-overlapping sequences |
| Priority exponent | 0.9 |
| Importance sampling exponent | 0 |
| Learning rule (downstream tasks) | $Q(\lambda)$, $\lambda = 0.7$ |
| Learning rule (NGU pre-training) | Retrace($\lambda$), $\lambda = 0.95$ |
| Discount (downstream tasks) | 0.99 |
| Discount (NGU pre-training) | 0.99 |
| Minibatch size | 64 |
| Optimizer | Adam |
| Optimizer settings | $\varepsilon = 10^{-4}$, $\beta_1 = 0.9$, $\beta_2 = 0.999$ |
| Learning rate | $2 \times 10^{-4}$ |
| Target network update interval | 1500 updates |
| $\epsilon_{\text{levy}}$ distribution | Log-Uniform$[0, 0.1]$ |
| Flight length distribution | Zeta with $\mu = 2$ |

It should be noted that our $\epsilon z$-greedy baseline under-performs relative to Dabney et al. [14]. This is due to our hyper-parameters and setting being derived from Puigdomènech Badia et al. [52], which adopts the standard Atari pre-processing (e.g. gray scale images and frame stacking). In contrast, Dabney et al. [14] use color images, no frame stacking, a larger neural network and different hyper-parameters (e.g. smaller replay buffer). Studying if the performance of NGU, RND and BT is preserved in this setting is an important direction for future work. We suspect that improving the performance of our $\epsilon z$-greedy ablation will also improve our method, since exploration flights are central to both.

## C Extended Unsupervised RL Results

We compare the results of our unsupervised pre-training stage against other unsupervised approaches, standard RL algorithms in the low-data regime and methods that perform unsupervised pre-training followed by an adaptation stage. Since the considered intrinsic rewards are non-negative, we consider a baseline where the agent obtains a constant positive reward at each step in order to measure the performance of policies that seek to stay alive for as long as possible. Results for this baseline were already considered by Hansen et al. [28] (*Pos Reward NSQ*), but we run our own version of this baseline using the distributed setting and longer pre-training of 16B frames considered in our experiments (*Pos Reward R2D2*). Table 2 shows that unsupervised RND and NGU outperform all baselines by a large margin, confirming the intuition that exploration is a good pre-training objective for the Atari benchmark. These results suggest that there is a strong correlation between exploration and the goals established by game designers [10]. In spite of the strong results, it is worth noting that unsupervised RND and NGU achieve lower scores than random policies in some games, and can be quite inefficient at collecting rewards in some environments (e.g. they needs long episodes to obtain high scores). These observations motivate the development of techniques to leverage these pre-trained policies without compromising performance even when there exists a misalignment between objectives.

Table 2: Atari Suite comparisons, adapted from Hansen et al. [28] and Liu and Abbeel [41]. @$N$ represents the amount of RL interaction with reward utilized, with four frames observed at each iteration. *Mdn* and *M* are median and mean human normalized scores, respectively; $> 0$ is the number of games with better than random performance; and $> H$ is the number of games with human-level performance as defined in Mnih et al. [43]. **Top**: unsupervised learning only. **Mid**: data-limited RL. **Bottom**: RL with unsupervised pre-training.

| Algorithm | 26 Game Subset Kaiser et al. [35] | | | | 47 Game Subset Burda et al. [10] | | | | Full 57 Games Mnih et al. [43] | | | |
|---|---|---|---|---|---|---|---|---|---|---|---|---|
| | Mdn | M | >0 | >H | Mdn | M | >0 | >H | Mdn | M | >0 | >H |
| IDF Curiosity @0 | – | – | – | – | 8.46 | 24.51 | 34 | 5 | – | – | – | – |
| RF Curiosity @0 | – | – | – | – | 7.32 | 29.03 | 36 | 6 | – | – | – | – |
| Pos Reward NSQ @0 | 2.18 | 50.33 | 14 | 5 | 0.69 | 57.65 | 26 | 8 | 0.29 | 41.19 | 28 | 8 |
| Pos Reward R2D2 @0 | 9.44 | 59.55 | 21 | 4 | 14.16 | 57.53 | 39 | 5 | 3.46 | 45.23 | 46 | 5 |
| Q-DIAYN-5 @0 | 0.17 | −3.60 | 13 | 0 | 0.33 | −1.23 | 25 | 2 | 0.34 | −2.18 | 30 | 2 |
| Q-DIAYN-50 @0 | −1.65 | −21.77 | 4 | 0 | −1.69 | −16.26 | 8 | 0 | −3.16 | −20.31 | 9 | 0 |
| VISR @0 | 5.60 | 81.65 | 19 | 5 | 4.04 | 58.47 | 35 | 7 | 3.77 | 49.66 | 40 | 7 |
| RND@0 | 48.35 | 334.65 | 23 | 8 | 41.28 | 259.43 | 40 | 14 | 40.86 | 243.01 | 47 | 16 |
| NGU @0 | 80.92 | **494.54** | 25 | 12 | **96.10** | **310.27** | 45 | 23 | **81.72** | **320.06** | **52** | **27** |
| SimPLe @100$k$ | 9.79 | 36.20 | 26 | 4 | – | – | – | – | – | – | – | – |
| DQN @10$M$ | 27.80 | 52.95 | 25 | 7 | 9.91 | 28.07 | 41 | 7 | 8.61 | 27.55 | 48 | 7 |
| DQN @200$M$ | **100.76** | 267.51 | **26** | **13** | – | – | – | – | 80.81 | 239.29 | 46 | 20 |
| Rainbow @100$k$ | 2.23 | 10.12 | 25 | 1 | – | – | – | – | – | – | – | – |
| PPO @500$k$ | 20.93 | 43.74 | 25 | 7 | – | – | – | – | – | – | – | – |
| NSQ @10$M$ | 8.20 | 33.80 | 22 | 3 | 7.29 | 29.47 | 37 | 4 | 6.80 | 28.51 | 43 | 5 |
| SPR @100$k$ | 41.50 | 70.40 | – | 7 | – | – | – | – | – | – | – | – |
| CURL @100$k$ | 17.50 | 38.10 | – | 2 | – | – | – | – | – | – | – | – |
| DrQ @100$k$ | 28.42 | 35.70 | – | 2 | – | – | – | – | – | – | – | – |
| Q-DIAYN-5 @100$k$ | 0.01 | 16.94 | 13 | 2 | 1.31 | 19.64 | 28 | 6 | 1.55 | 16.65 | 33 | 6 |
| Q-DIAYN-50 @100$k$ | −1.64 | −27.88 | 3 | 0 | −1.66 | −16.74 | 8 | 0 | −2.53 | −24.13 | 9 | 0 |
| RF VISR @100$k$ | 7.24 | 58.23 | 20 | 6 | 3.81 | 42.60 | 33 | 9 | 2.16 | 35.29 | 39 | 9 |
| VISR @100$k$ | 9.50 | 128.07 | 21 | 7 | 9.42 | 121.08 | 35 | 11 | 6.81 | 102.31 | 40 | 11 |
| GPI RF VISR @100$k$ | 5.55 | 58.77 | 20 | 5 | 4.24 | 48.38 | 34 | 9 | 3.60 | 40.01 | 40 | 10 |
| GPI VISR @100$k$ | 6.59 | 111.23 | 22 | 7 | 11.70 | 129.76 | 38 | 12 | 8.99 | 109.16 | 44 | 12 |
| MEPOL @100$k$ | 0.34 | 17.94 | – | 2 | – | – | – | – | – | – | – | – |
| APT @100$k$ | 47.50 | 69.55 | – | 7 | – | – | – | – | 33.41 | 47.73 | – | 12 |

# D  Extended Atari-57 Results

Table 3: Atari Suite comparisons for R2D2-based agents. @$N$ represents the amount of frames with reward utilized, with four frames observed per RL interaction. *Mdn*, *M* and *CM* are median, mean and mean capped human normalized scores, respectively.

| Algorithm | Full 57 Games | | | Hard Exploration | | |
|---|---|---|---|---|---|---|
| | Mdn | M | CM | Mdn | M | CM |
| R2D2 @$1B$ | 229.75 | 864.69 | 84.56 | 31.07 | 39.40 | 34.75 |
| R2D2 + $\epsilon z$-greedy @$1B$ | 204.52 | 578.73 | 85.11 | 42.55 | 53.90 | 46.21 |
| R2D2 + BT($\pi_{\text{NGU}}$) @$1B$ | 273.49 | **1517.13** | 86.38 | **100.89** | **94.20** | 63.95 |
| R2D2 + BT($\pi_{\text{RND}}$) @$1B$ | **280.04** | 1396.78 | **87.43** | 93.52 | 86.75 | **67.40** |
| R2D2 @$5B$ | 490.12 | 1742.92 | 90.37 | 32.49 | 67.41 | 44.74 |
| R2D2 + $\epsilon z$-greedy @$5B$ | 418.41 | 1275.86 | 92.49 | 103.62 | 95.46 | 67.85 |
| R2D2 + BT($\pi_{\text{NGU}}$) @$5B$ | 538.50 | 2262.21 | **93.31** | **193.15** | **160.02** | 76.92 |
| R2D2 + BT($\pi_{\text{RND}}$) @$5B$ | **571.57** | **2304.19** | 92.03 | 144.78 | 123.38 | **76.93** |

Table 4: Atari Suite comparisons with rewards for R2D2-based agents with different amounts of transfer via weights at 5B training frames. Policies are composed of a CNN encoder followed by an LSTM and a dueling head. We compare training from scratch, loading all weights (Full $\pi_{\text{NGU}}$ init) or all weights except those in the dueling head (Partial $\pi_{\text{NGU}}$ init). *Mdn*, *M* and *CM* are median, mean and mean capped human normalized scores, respectively. (**Top**) Without BT. (**Bottom**) With BT($\pi_{\text{NGU}}$).

| Algorithm | Full 57 Games | | | Hard Exploration | | |
|---|---|---|---|---|---|---|
| | Mdn | M | CM | Mdn | M | CM |
| R2D2, from scratch | 490.12 | 1742.92 | 90.37 | 32.49 | 67.41 | 44.74 |
| R2D2, partial $\pi_{\text{NGU}}$ init | **668.80** | 2020.81 | **93.00** | **109.33** | **123.40** | **67.18** |
| R2D2, full $\pi_{\text{NGU}}$ init | 507.58 | **2359.25** | 89.91 | 104.98 | 101.52 | 66.20 |
| R2D2 + BT($\pi_{\text{NGU}}$), from scratch | 538.50 | 2262.21 | 93.31 | 193.15 | 160.02 | 76.92 |
| R2D2 + BT($\pi_{\text{NGU}}$), partial $\pi_{\text{NGU}}$ init | **626.34** | 1966.83 | **94.07** | **200.32** | **164.54** | **76.93** |
| R2D2 + BT($\pi_{\text{NGU}}$), full $\pi_{\text{NGU}}$ init | 529.78 | **2467.02** | 92.79 | 168.18 | 137.65 | **76.93** |

Table 5: Human normalized scores after 5B frames with rewards for R2D2-based agents at different percentiles. Note that the 50th percentile corresponds to the median score across the 57 games. We compare training from scratch, loading all weights (Full $\pi_{\text{NGU}}$ init) or all weights except those in the dueling head (Partial $\pi_{\text{NGU}}$ init).

| Method | Percentile | | | | |
|---|---|---|---|---|---|
| | 50th | 40th | 20th | 10th | 5th |
| R2D2, from scratch | 490.12 | 220.97 | 132.77 | 92.31 | 25.57 |
| R2D2 + BT($\pi_{\text{NGU}}$), from scratch | 538.50 | **316.30** | **163.20** | **104.41** | **65.17** |
| R2D2 + BT($\pi_{\text{RND}}$), from scratch | **571.57** | 279.97 | 133.61 | 87.05 | 54.27 |
| R2D2, partial $\pi_{\text{NGU}}$ init | **668.80** | 434.02 | 164.62 | 105.08 | 53.59 |
| R2D2 + BT($\pi_{\text{NGU}}$), partial $\pi_{\text{NGU}}$ init | 626.34 | **489.14** | **171.69** | **113.92** | **93.52** |
| R2D2, full $\pi_{\text{NGU}}$ init | 507.58 | 293.54 | 137.79 | 59.72 | 41.06 |
| R2D2 + BT($\pi_{\text{NGU}}$), full $\pi_{\text{NGU}}$ init | **529.78** | **384.39** | **163.55** | **123.56** | **51.98** |

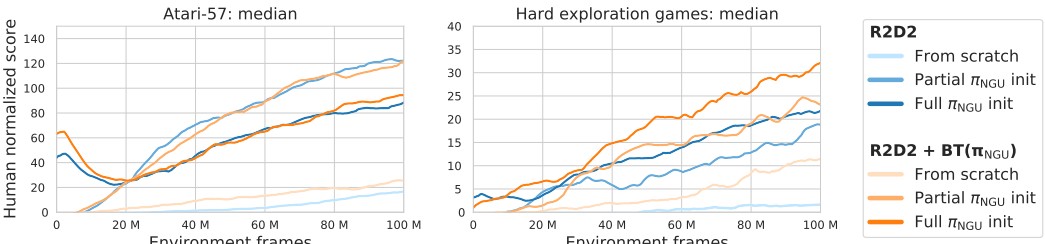

Figure 8: Median human normalized scores for R2D2-based agents with different amounts of transfer via weights during the first 100M frames of training. Policies are composed of a CNN encoder followed by an LSTM and a dueling head. We compare training from scratch, loading the CNN and the LSTM (Partial $\pi_{\text{NGU}}$ init), and loading all weights including the dueling head (Full $\pi_{\text{NGU}}$ init). (**Left**) Full Atari suite. (**Right**) Subset of hard exploration games.

## E  Alternative Reward Functions

**MsPacman: eating ghosts**

- Pac-dots: 0 points (easy) or -10 points (hard)
- Eating vulnerable ghosts:
  - #1 in succession: 200 points
  - #2 in succession: 400 points
  - #3 in succession: 800 points
  - #4 in succession: 1600 points
- Other actions: 0 points

**Hero: rescuing miners**

- Dynamiting walls: 0 points (easy) or -300 points (hard)
- Rescuing a miner: 1000 points
- Other actions: 0 points

## F  Distributed setting

All experiments are run using a distributed setting. The evaluation we do is also identical to the one done in R2D2 [37]: parallel evaluation workers, which share weights with actors and learners, run the Q-network against the environment. This worker and all the actor workers are the two types of workers that draw samples from the environment. For Atari, we apply the standard DQN pre-processing, as used in R2D2. The next subsections describe how actors, evaluators, and learner are run in each stage.

### F.1  Unsupervised stage

The computation of the intrinsic NGU reward, $r_t^{\text{NGU}}$, follows the method described in Puig-domènech Badia et al. [52, Appendix A.1]. In particular, we use the version that combines episodic intrinsic rewards with the intrinsic reward from Random Network Distillation (RND) [11].

We now describe the distributed setup used for NGU, which is largely the same as the one used for RND. Note that RND can be recovered by removing the components needed for the episodic reward.

**Learner**

- Sample from the replay buffer a sequence of intrinsic rewards $r_t^{\text{NGU}}$, observations $x$ and actions $a$.
- Use Q-network to learn from $(r_t^{\text{NGU}}, x, a)$ with Retrace [45] using the same procedure as in R2D2.

- Use last 5 frames of the sampled sequences to train the action prediction network in NGU. This means that, for every batch of sequences, all time steps are used to train the RL loss, whereas only 5 time steps per sequence are used to optimize the action prediction loss.

- Use last 5 frames of the sampled sequences to train the predictor of RND.

**Actor**

- Obtain $x_t$ and $r_{t-1}^{\text{NGU}}$.

- With these inputs, compute forward pass of R2D2 to obtain $a_t$.

- With $x_t$, compute $r_t^{\text{NGU}}$ using the embedding network in NGU.

- Insert $x_t$, $a_t$ and $r_t^{\text{NGU}}$ in the replay buffer.

- Step on the environment with $a_t$.

**Evaluator**

- Obtain $x_t$ and $r_{t-1}^{\text{NGU}}$.

- With these inputs, compute forward pass of R2D2 to obtain $a_t$.

- With $x_t$, compute $r_t^{\text{NGU}}$ using the embedding network in NGU.

- Step on the environment with $a_t$.

**Distributed training**

As in R2D2, we train the agent with a single GPU-based learner and a fixed discount factor $\gamma$. All actors collect experience using the same policy, but with a different value of $\epsilon$. This differs from the original NGU agent, where each actor runs a policy with a different degree of exploratory behavior and discount factor.

In the replay buffer, we store fixed-length sequences of $(x, a, r)$ tuples. These sequences never cross episode boundaries. Given a single batch of trajectories we unroll both online and target networks on the same sequence of states to generate value estimates. We use prioritized experience replay with the same prioritization scheme proposed in [37].

## F.2   Transfer with BT

**Learner**

- Sample from the replay buffer a sequence of extrinsic rewards $r_t$, observations $x$ and actions $a$.

- (expanded action set) Duplicate transitions collected with $\pi_p$ and relabel the duplicates with the primitive action taken by $\pi_p$ when acting.

- Use Q-network to learn from $(r_t, x, a)$ with Peng's Q($\lambda$) [50] using the same procedure as in R2D2.

**Actor**

- (once per episode) Sample $\epsilon_{\text{levy}}$.

- Obtain $x_t$.

- If not on a flight, start one with probability $\epsilon_{\text{levy}}$.

- If on a flight, compute forward pass with $\pi_p$ to obtain $a_t$. Otherwise, compute forward pass of R2D2 to obtain $a_t$. If $a_t = |\mathcal{A}| + 1$, $a_t \leftarrow \pi_p(x)$.

- Insert $x_t$, $a_t$ and $r_t$ in the replay buffer.

- Step on the environment with $a_t$.

**Evaluator**

- Obtain $x_t$.
- Compute forward pass of R2D2 to obtain $a_t$. If $a_t = |\mathcal{A}| + 1$, $a_t \leftarrow \pi_p(x)$.
- Step on the environment with $a_t$.

**Distributed training**

As in R2D2, we train the agent with a single GPU-based learner and a fixed discount factor $\gamma$. All actors collect experience using the same policy, but with a different value of $\epsilon$.

In the replay buffer, we store fixed-length sequences of $(x, a, r)$ tuples. These sequences never cross episode boundaries. Given a single batch of trajectories we unroll both online and target networks on the same sequence of states to generate value estimates. We use prioritized experience replay with the same prioritization scheme proposed in [37].

# G  Intrinsic Rewards

## G.1  Random Network Distillation

The RND [11] intrinsic reward is computed by introducing a random, untrained convolutional network $g : \mathcal{S} \to \mathbb{R}^d$, and training a network $\hat{g} : \mathcal{S} \to \mathbb{R}^d$ to predict the outputs of $g$ on all the observations that are seen during training by minimizing the prediction error $\text{err}_{\text{RND}}(s_t) = ||\hat{g}(s_t; \theta) - g(s_t)||^2$ with respect to $\theta$. The intuition is that the prediction error will be large on states that have been visited less frequently by the agent. The dimensionality of the random embedding, $d$, is a hyperparameter of the algorithm.

The RND intrinsic reward is obtained by normalising the prediction error. In this work, we use a slightly different normalization from that reported in [11]. The RND reward at time $t$ is given by

$$r_t^{\text{RND}} = \frac{\text{err}_{\text{RND}}(s_t)}{\sigma_e} \tag{2}$$

where $\sigma_e$ is the running standard deviation of $\text{err}_{\text{RND}}(s_t)$.

## G.2  Never Give Up

The NGU intrinsic reward modulates an episodic intrinsic reward, $r_t^{\text{episodic}}$, with a life long signal $\alpha_t$:

$$r_t^{\text{NGU}} = r_t^{\text{episodic}} \cdot \min\left\{\max\left\{\alpha_t, 1\right\}, L\right\}, \tag{3}$$

where $L$ is a fixed maximum reward scaling. The life-long novelty signal is computed using RND with the normalisation:

$$\alpha_t = \frac{\text{err}_{\text{RND}}(s_t) - \mu_e}{\sigma_e} \tag{4}$$

where $\text{err}_{\text{RND}}(x_t)$ is the prediction error described in Appendix G.1, and $\mu_e$ and $\sigma_e$ are its running mean and standard deviation, respectively. The episodic intrinsic reward at time $t$ is computed according to formula:

$$r_t^{\text{episodic}} = \frac{1}{\sqrt{\sum_{f(s_i) \in N_k} K(f(s_t), f(s_i))} + c} \tag{5}$$

where $N_k$ is the set containing the $k$-nearest neighbors of $f(s_t)$ in $M$, $c$ is a constant and $K : \mathbb{R}^p \times \mathbb{R}^p \to \mathbb{R}^+$ is a kernel function satisfying $K(x, x) = 1$ (which can be thought of as approximating pseudo-counts [52]). Algorithm 4 shows a detailed description of how the episodic intrinsic reward is computed. Below we describe the different components used in Algorithm 4:

- $M$: episodic memory containing at time $t$ the previous embeddings $\{f(s_0), f(s_1), \ldots, f(s_{t-1})\}$. This memory starts empty at each episode
- $k$: number of nearest neighbours

- $N_k = \{f(s_i)\}_{i=1}^k$: set of $k$-nearest neighbours of $f(s_t)$ in the memory $M$; we call $N_k[i] = f(s_i) \in N_k$ for ease of notation
- $K$: kernel defined as $K(x,y) = \frac{\epsilon}{\frac{d^2(x,y)}{d_m^2} + \epsilon}$ where $\epsilon$ is a small constant, $d$ is the Euclidean distance and $d_m^2$ is a running average of the squared Euclidean distance of the $k$-nearest neighbors
- $c$: pseudo-counts constant
- $\xi$: cluster distance
- $s_m$: maximum similarity

---

**Algorithm 4:** Computation of the episodic intrinsic reward at time $t$: $r_t^{\text{episodic}}$.

---

**Input** : $M$; $k$; $f(s_t)$; $c$; $\epsilon$; $\xi$; $s_m$; $d_m^2$
**Output :** $r_t^{\text{episodic}}$

Compute the $k$-nearest neighbours of $f(s_t)$ in $M$ and store them in a list $N_k$
Create a list of floats $d_k$ of size $k$
/* The list $d_k$ will contain the distances between the embedding $f(s_t)$ and
   its neighbours $N_k$.                */
**for** $i \in \{1, \ldots, k\}$ **do**
   $d_k[i] \leftarrow d^2(f(s_t), N_k[i])$
**end**
Update the moving average $d_m^2$ with the list of distances $d_k$
/* Normalize the distances $d_k$ with the updated moving average $d_m^2$.   */
$d_n \leftarrow \frac{d_k}{d_m^2}$
/* Cluster the normalized distances $d_n$ i.e. they become 0 if too small
   and $0_k$ is a list of $k$ zeros.               */
$d_n \leftarrow \max(d_n - \xi, 0_k)$
/* Compute the Kernel values between the embedding $f(s_t)$ and its
   neighbours $N_k$.               */
$K_v \leftarrow \frac{\epsilon}{d_n + \epsilon}$
/* Compute the similarity between the embedding $f(s_t)$ and its neighbours
   $N_k$.               */
$s \leftarrow \sqrt{\sum_{i=1}^k K_v[i]} + c$
/* Compute the episodic intrinsic reward at time $t$: $r_t^i$.      */
**if** $s > s_m$ **then**
   $r_t^{\text{episodic}} \leftarrow 0$
**else**
   $r_t^{\text{episodic}} \leftarrow 1/s$

 # H Scores per game

Table 6: Results per game for R2D2-based agents at 5B training frames.

| Game | R2D2 | R2D2 + $\epsilon z$-greedy | R2D2 + BT($\pi_{\text{NGU}}$) | R2D2 + BT($\pi_{\text{RND}}$) |
|---|---|---|---|---|
| alien | $10831.17 \pm 2114.29$ | $14634.02 \pm 1109.15$ | $\mathbf{15657.57 \pm 1717.96}$ | $12844.24 \pm 1447.72$ |
| amidar | $\mathbf{11761.67 \pm 1560.86}$ | $6784.28 \pm 718.05$ | $10394.96 \pm 891.60$ | $7730.43 \pm 670.76$ |
| assault | $\mathbf{15940.72 \pm 3531.69}$ | $9177.28 \pm 2170.26$ | $15060.31 \pm 740.63$ | $11533.24 \pm 809.49$ |
| asterix | $472812.21 \pm 222663.81$ | $374966.62 \pm 135810.51$ | $\mathbf{630663.91 \pm 82753.46}$ | $468724.08 \pm 120822.86$ |
| asteroids | $45716.28 \pm 3642.38$ | $\mathbf{147005.85 \pm 44313.45}$ | $31957.42 \pm 15540.09$ | $37455.64 \pm 9263.04$ |
| atlantis | $1514724.43 \pm 10941.36$ | $1132188.04 \pm 43551.36$ | $1491384.23 \pm 5978.05$ | $\mathbf{1545954.35 \pm 9001.60}$ |
| bank heist | $965.63 \pm 133.72$ | $1058.75 \pm 135.46$ | $13913.32 \pm 3529.15$ | $82132.27 \pm 101709.64$ |
| battle zone | $292553.41 \pm 18196.77$ | $\mathbf{312367.76 \pm 43554.18}$ | $258533.57 \pm 22865.64$ | $285925.87 \pm 44912.86$ |
| beam rider | $18472.45 \pm 1977.78$ | $\mathbf{22403.95 \pm 1596.92}$ | $16301.02 \pm 1853.73$ | $15619.99 \pm 2048.77$ |
| berzerk | $12343.83 \pm 3331.54$ | $3846.56 \pm 1723.24$ | $8359.80 \pm 201.10$ | $\mathbf{14687.68 \pm 401.76}$ |
| bowling | $141.64 \pm 4.52$ | $156.32 \pm 8.11$ | $174.27 \pm 0.10$ | $\mathbf{196.01 \pm 57.42}$ |
| boxing | $99.96 \pm 0.03$ | $99.94 \pm 0.06$ | $\mathbf{100.00 \pm 0.00}$ | $99.98 \pm 0.03$ |
| breakout | $432.65 \pm 27.35$ | $393.19 \pm 35.12$ | $\mathbf{441.21 \pm 15.08}$ | $429.38 \pm 15.52$ |
| centipede | $189502.66 \pm 31388.08$ | $\mathbf{358841.20 \pm 73578.20}$ | $178635.17 \pm 17227.15$ | $196880.46 \pm 24278.18$ |
| chopper command | $611393.11 \pm 65206.69$ | $697655.53 \pm 215090.74$ | $573055.88 \pm 75343.57$ | $\mathbf{797052.58 \pm 52012.04}$ |
| crazy climber | $\mathbf{229992.57 \pm 17738.33}$ | $212001.76 \pm 1853.07$ | $226821.26 \pm 3608.19$ | $198736.17 \pm 7631.83$ |
| defender | $\mathbf{547238.15 \pm 2579.38}$ | $516521.06 \pm 11969.59$ | $540124.74 \pm 4488.40$ | $524003.44 \pm 1316.59$ |
| demon attack | $143662.42 \pm 88.16$ | $141352.18 \pm 3848.73$ | $\mathbf{143762.91 \pm 106.75}$ | $143578.47 \pm 25.05$ |
| double dunk | $\mathbf{23.99 \pm 0.02}$ | $23.88 \pm 0.06$ | $23.85 \pm 0.15$ | $23.93 \pm 0.05$ |
| enduro | $2358.37 \pm 3.32$ | $2359.08 \pm 1.03$ | $\mathbf{2361.56 \pm 1.03}$ | $2350.39 \pm 8.42$ |
| fishing derby | $12.80 \pm 77.79$ | $\mathbf{64.74 \pm 0.59}$ | $52.58 \pm 0.32$ | $62.11 \pm 5.59$ |
| freeway | $\mathbf{33.87 \pm 0.08}$ | $33.77 \pm 0.03$ | $33.79 \pm 0.08$ | $33.79 \pm 0.07$ |
| frostbite | $9287.24 \pm 167.11$ | $8504.41 \pm 940.72$ | $\mathbf{17692.42 \pm 2871.83}$ | $9419.45 \pm 188.92$ |
| gopher | $\mathbf{117398.58 \pm 2485.82}$ | $84140.40 \pm 12919.83$ | $113716.78 \pm 3966.91$ | $94670.35 \pm 2285.63$ |
| gravitar | $6123.08 \pm 103.19$ | $5798.68 \pm 735.59$ | $\mathbf{8373.70 \pm 1260.75}$ | $7428.57 \pm 2459.91$ |
| hero | $\mathbf{46048.07 \pm 6970.26}$ | $39700.22 \pm 4379.84$ | $40825.09 \pm 3736.26$ | $42959.86 \pm 7950.56$ |
| ice hockey | $32.43 \pm 30.64$ | $30.65 \pm 28.17$ | $\mathbf{60.36 \pm 4.94}$ | $57.96 \pm 0.90$ |
| jamesbond | $\mathbf{6056.14 \pm 1643.52}$ | $3843.92 \pm 118.35$ | $1484.87 \pm 489.66$ | $2870.03 \pm 907.76$ |
| kangaroo | $14672.37 \pm 187.16$ | $14730.99 \pm 114.20$ | $\mathbf{15965.79 \pm 36.61}$ | $15128.66 \pm 188.17$ |
| krull | $10081.04 \pm 594.10$ | $10171.52 \pm 399.81$ | $\mathbf{406596.00 \pm 55547.76}$ | $316960.78 \pm 217091.10$ |
| kung fu master | $200721.64 \pm 2265.35$ | $171591.29 \pm 8516.87$ | $196638.89 \pm 456.09$ | $\mathbf{610699.23 \pm 60053.99}$ |
| montezuma revenge | $1478.38 \pm 1114.20$ | $1467.77 \pm 1104.72$ | $\mathbf{12086.71 \pm 1217.76}$ | $6266.67 \pm 471.40$ |
| ms pacman | $\mathbf{11212.85 \pm 103.23}$ | $7511.39 \pm 406.77$ | $10996.90 \pm 262.74$ | $10656.00 \pm 356.46$ |
| name this game | $32138.12 \pm 2156.95$ | $\mathbf{37343.04 \pm 1917.73}$ | $30252.11 \pm 884.84$ | $28746.14 \pm 1798.77$ |
| phoenix | $\mathbf{712101.72 \pm 62738.09}$ | $80611.18 \pm 25316.56$ | $553429.34 \pm 24278.55$ | $283686.99 \pm 172323.63$ |
| pitfall | $-0.19 \pm 0.15$ | $-12.34 \pm 4.20$ | $-0.39 \pm 0.39$ | $\mathbf{-0.03 \pm 0.04}$ |
| pong | $20.93 \pm 0.01$ | $20.49 \pm 0.10$ | $20.90 \pm 0.01$ | $\mathbf{20.94 \pm 0.01}$ |
| private eye | $23592.22 \pm 11876.55$ | $\mathbf{50770.82 \pm 14984.92}$ | $40435.54 \pm 51.04$ | $40480.67 \pm 38.23$ |
| qbert | $\mathbf{24343.75 \pm 1904.89}$ | $16975.13 \pm 1332.44$ | $16057.31 \pm 318.87$ | $10990.08 \pm 7241.50$ |
| riverraid | $\mathbf{32325.07 \pm 1185.15}$ | $30582.53 \pm 638.47$ | $28550.32 \pm 2298.03$ | $30566.86 \pm 1764.50$ |
| road runner | $\mathbf{423191.07 \pm 53071.15}$ | $88890.04 \pm 24971.18$ | $251261.09 \pm 31741.38$ | $248661.22 \pm 19416.63$ |
| robotank | $97.23 \pm 1.22$ | $\mathbf{108.92 \pm 4.79}$ | $98.45 \pm 2.85$ | $100.57 \pm 6.32$ |
| seaquest | $\mathbf{188771.84 \pm 20759.57}$ | $175745.09 \pm 120718.82$ | $86605.86 \pm 55065.85$ | $38185.98 \pm 22949.18$ |
| skiing | $-29854.11 \pm 85.79$ | $-30060.81 \pm 142.32$ | $-30121.95 \pm 70.62$ | $\mathbf{-29589.38 \pm 69.40}$ |
| solaris | $17741.02 \pm 5340.46$ | $16127.73 \pm 2975.20$ | $\mathbf{24366.59 \pm 4868.05}$ | $18727.45 \pm 4806.17$ |
| space invaders | $3621.76 \pm 5.81$ | $3547.78 \pm 35.31$ | $30609.21 \pm 7141.11$ | $\mathbf{46704.49 \pm 7017.79}$ |
| star gunner | $\mathbf{223536.63 \pm 48548.34}$ | $179698.69 \pm 12194.36$ | $171294.31 \pm 23185.79$ | $156691.39 \pm 16704.19$ |
| surround | $\mathbf{8.24 \pm 0.48}$ | $1.48 \pm 8.12$ | $5.86 \pm 1.44$ | $-3.62 \pm 4.79$ |
| tennis | $7.99 \pm 22.56$ | $7.98 \pm 22.51$ | $\mathbf{23.96 \pm 0.01}$ | $7.97 \pm 22.56$ |
| time pilot | $\mathbf{139931.67 \pm 70521.78}$ | $71768.84 \pm 2933.22$ | $44936.87 \pm 137.49$ | $77711.97 \pm 4735.53$ |
| tutankham | $324.02 \pm 4.26$ | $311.65 \pm 8.62$ | $\mathbf{420.36 \pm 30.13}$ | $357.26 \pm 14.22$ |
| up n down | $529363.05 \pm 16813.20$ | $394984.70 \pm 34313.42$ | $562739.02 \pm 8527.59$ | $\mathbf{585355.01 \pm 4718.67}$ |
| venture | $0.00 \pm 0.00$ | $1833.85 \pm 43.73$ | $\mathbf{2110.64 \pm 55.39}$ | $1910.15 \pm 13.98$ |
| video pinball | $454023.46 \pm 377076.03$ | $107071.98 \pm 67142.18$ | $463141.28 \pm 426927.92$ | $\mathbf{646671.78 \pm 403584.41}$ |
| wizard of wor | $\mathbf{40833.65 \pm 4776.81}$ | $38275.31 \pm 4177.41$ | $30453.12 \pm 4770.20$ | $30399.63 \pm 2345.83$ |
| yars revenge | $279765.86 \pm 27370.20$ | $250483.70 \pm 54593.32$ | $\mathbf{280333.48 \pm 69704.31}$ | $200850.59 \pm 72885.67$ |
| zaxxon | $56059.14 \pm 3217.77$ | $66099.28 \pm 8520.19$ | $\mathbf{67611.78 \pm 6226.04}$ | $59926.08 \pm 5834.47$ |

Table 7: Results per game for R2D2 agents with different amounts of transfer via weights at 5B training frames. Policies are composed of a CNN encoder followed by an LSTM and a dueling head. We compare training from scratch, loading all weights (Full $\pi_{\mathrm{NGU}}$ init) or all weights except those in the dueling head (Partial $\pi_{\mathrm{NGU}}$ init).

| Game | From scratch | Partial $\pi_{\mathrm{NGU}}$ init | Full $\pi_{\mathrm{NGU}}$ init |
|---|---|---|---|
| alien | $10831.17 \pm 2114.29$ | $\mathbf{27299.78 \pm 5730.57}$ | $18027.35 \pm 6731.75$ |
| amidar | $11761.67 \pm 1560.86$ | $\mathbf{13647.09 \pm 3380.90}$ | $3518.30 \pm 2353.96$ |
| assault | $\mathbf{15940.72 \pm 3531.69}$ | $14653.32 \pm 2047.43$ | $12533.61 \pm 1001.68$ |
| asterix | $472812.21 \pm 222663.81$ | $\mathbf{789344.47 \pm 80638.00}$ | $676662.54 \pm 8536.94$ |
| asteroids | $45716.28 \pm 3642.38$ | $\mathbf{73298.12 \pm 22688.38}$ | $23127.43 \pm 5425.84$ |
| atlantis | $1514724.43 \pm 10941.36$ | $1537659.81 \pm 7693.86$ | $\mathbf{1556234.51 \pm 9709.74}$ |
| bank heist | $965.63 \pm 133.72$ | $1841.77 \pm 52.75$ | $\mathbf{5816.24 \pm 3137.60}$ |
| battle zone | $292553.41 \pm 18196.77$ | $\mathbf{301715.60 \pm 12875.97}$ | $248939.89 \pm 31788.00$ |
| beam rider | $\mathbf{18472.45 \pm 1977.78}$ | $16179.19 \pm 4179.36$ | $11040.66 \pm 799.77$ |
| berzerk | $12343.83 \pm 3331.54$ | $16888.63 \pm 2330.72$ | $\mathbf{25465.10 \pm 9886.89}$ |
| bowling | $141.64 \pm 4.52$ | $170.36 \pm 16.55$ | $\mathbf{180.78 \pm 2.94}$ |
| boxing | $99.96 \pm 0.03$ | $\mathbf{99.97 \pm 0.05}$ | $99.94 \pm 0.07$ |
| breakout | $432.65 \pm 27.35$ | $\mathbf{520.19 \pm 64.65}$ | $487.51 \pm 49.14$ |
| centipede | $189502.66 \pm 31388.08$ | $\mathbf{528000.27 \pm 10403.62}$ | $500534.38 \pm 9267.05$ |
| chopper command | $611393.11 \pm 65206.69$ | $\mathbf{937637.69 \pm 57836.24}$ | $764150.71 \pm 44756.44$ |
| crazy climber | $229992.57 \pm 17738.33$ | $\mathbf{275735.70 \pm 15244.51}$ | $246498.24 \pm 12319.58$ |
| defender | $\mathbf{547238.15 \pm 2579.38}$ | $534656.86 \pm 2880.53$ | $523660.11 \pm 2604.26$ |
| demon attack | $\mathbf{143662.42 \pm 88.16}$ | $143592.16 \pm 77.27$ | $143574.75 \pm 69.35$ |
| double dunk | $\mathbf{23.99 \pm 0.02}$ | $\mathbf{23.99 \pm 0.02}$ | $23.83 \pm 0.06$ |
| enduro | $2358.37 \pm 3.32$ | $\mathbf{2359.39 \pm 8.12}$ | $2353.16 \pm 1.30$ |
| fishing derby | $12.80 \pm 77.79$ | $\mathbf{68.70 \pm 2.46}$ | $59.22 \pm 2.55$ |
| freeway | $\mathbf{33.87 \pm 0.08}$ | $33.83 \pm 0.06$ | $33.79 \pm 0.04$ |
| frostbite | $9287.24 \pm 167.11$ | $\mathbf{161595.33 \pm 32917.44}$ | $10307.96 \pm 1087.09$ |
| gopher | $\mathbf{117398.58 \pm 2485.82}$ | $113094.41 \pm 4837.16$ | $102781.75 \pm 9613.77$ |
| gravitar | $6123.08 \pm 103.19$ | $\mathbf{7090.19 \pm 1359.52}$ | $5174.90 \pm 544.76$ |
| hero | $\mathbf{46048.07 \pm 6970.26}$ | $43982.29 \pm 4124.79$ | $40628.07 \pm 4008.99$ |
| ice hockey | $32.43 \pm 30.64$ | $\mathbf{69.57 \pm 1.18}$ | $47.67 \pm 10.59$ |
| jamesbond | $6056.14 \pm 1643.52$ | $\mathbf{6109.60 \pm 1643.75}$ | $3979.12 \pm 1233.92$ |
| kangaroo | $14672.37 \pm 187.16$ | $14863.32 \pm 259.85$ | $\mathbf{15192.97 \pm 832.48}$ |
| krull | $10081.04 \pm 594.10$ | $11806.49 \pm 580.05$ | $\mathbf{372307.71 \pm 161921.43}$ |
| kung fu master | $200721.64 \pm 2265.35$ | $200305.15 \pm 5711.26$ | $\mathbf{207401.69 \pm 1755.69}$ |
| montezuma revenge | $1478.38 \pm 1114.20$ | $\mathbf{2666.30 \pm 235.18}$ | $2500.00 \pm 0.00$ |
| ms pacman | $11212.85 \pm 103.23$ | $\mathbf{11795.03 \pm 640.73}$ | $11509.67 \pm 563.98$ |
| name this game | $32138.12 \pm 2156.95$ | $\mathbf{33811.87 \pm 2091.30}$ | $29242.89 \pm 1113.73$ |
| phoenix | $712101.72 \pm 62738.09$ | $\mathbf{812093.31 \pm 42328.98}$ | $801952.54 \pm 33211.40$ |
| pitfall | $\mathbf{-0.19 \pm 0.15}$ | $-1.43 \pm 1.17$ | $-0.61 \pm 0.60$ |
| pong | $20.93 \pm 0.01$ | $\mathbf{20.96 \pm 0.01}$ | $20.79 \pm 0.13$ |
| private eye | $23592.22 \pm 11876.55$ | $\mathbf{30345.57 \pm 10971.52}$ | $28653.02 \pm 9512.72$ |
| qbert | $24343.75 \pm 1904.89$ | $40943.28 \pm 16722.72$ | $\mathbf{62018.46 \pm 34865.08}$ |
| riverraid | $32325.07 \pm 1185.15$ | $\mathbf{35995.19 \pm 825.70}$ | $35845.18 \pm 3486.49$ |
| road runner | $\mathbf{423191.07 \pm 53071.15}$ | $311557.84 \pm 59675.36$ | $279988.24 \pm 58503.61$ |
| robotank | $97.23 \pm 1.22$ | $\mathbf{111.78 \pm 4.63}$ | $91.93 \pm 2.09$ |
| seaquest | $188771.84 \pm 20759.57$ | $\mathbf{629817.31 \pm 145648.54}$ | $31735.24 \pm 31257.98$ |
| skiing | $-29854.11 \pm 85.79$ | $\mathbf{-29550.10 \pm 495.22}$ | $-29981.62 \pm 564.13$ |
| solaris | $17741.02 \pm 5340.46$ | $\mathbf{29751.08 \pm 2076.41}$ | $22269.53 \pm 6584.51$ |
| space invaders | $3621.76 \pm 5.81$ | $41357.74 \pm 8968.52$ | $\mathbf{42695.22 \pm 7148.08}$ |
| star gunner | $\mathbf{223536.63 \pm 48548.34}$ | $212821.27 \pm 19723.78$ | $129058.91 \pm 11260.80$ |
| surround | $\mathbf{8.24 \pm 0.48}$ | $6.86 \pm 0.27$ | $-3.29 \pm 8.36$ |
| tennis | $7.99 \pm 22.56$ | $\mathbf{23.93 \pm 0.02}$ | $23.74 \pm 0.16$ |
| time pilot | $\mathbf{139931.67 \pm 70521.78}$ | $65101.20 \pm 7622.18$ | $49957.68 \pm 602.83$ |
| tutankham | $324.02 \pm 4.26$ | $\mathbf{333.37 \pm 10.62}$ | $312.19 \pm 4.32$ |
| up n down | $529363.05 \pm 16813.20$ | $572472.73 \pm 4512.30$ | $\mathbf{595047.70 \pm 1976.45}$ |
| venture | $0.00 \pm 0.00$ | $1930.36 \pm 32.97$ | $\mathbf{1958.13 \pm 48.98}$ |
| video pinball | $\mathbf{454023.46 \pm 377076.03}$ | $113036.77 \pm 3633.15$ | $108849.58 \pm 4753.59$ |
| wizard of wor | $40833.65 \pm 4776.81$ | $\mathbf{46931.25 \pm 1708.52}$ | $19100.00 \pm 1930.29$ |
| yars revenge | $279765.86 \pm 27370.20$ | $284565.01 \pm 29764.09$ | $\mathbf{294877.82 \pm 52551.36}$ |
| zaxxon | $56059.14 \pm 3217.77$ | $\mathbf{77649.92 \pm 15901.03}$ | $75850.01 \pm 10805.13$ |

Table 8: Results per game for R2D2+BT($\pi_{\text{NGU}}$) agents with different amounts of transfer via weights at 5B training frames. Policies are composed of a CNN encoder followed by an LSTM and a dueling head. We compare training from scratch, loading all weights (Full $\pi_{\text{NGU}}$ init) or all weights except those in the dueling head (Partial $\pi_{\text{NGU}}$ init).

| Game | From scratch | Partial $\pi_{\text{NGU}}$ init | Full $\pi_{\text{NGU}}$ init |
|---|---|---|---|
| alien | $15657.57 \pm 1717.96$ | $\mathbf{35441.47 \pm 3848.23}$ | $32822.74 \pm 3181.51$ |
| amidar | $10394.96 \pm 891.60$ | $\mathbf{11564.75 \pm 1726.50}$ | $8185.74 \pm 346.35$ |
| assault | $\mathbf{15060.31 \pm 740.63}$ | $12617.35 \pm 3267.80$ | $12992.70 \pm 2783.13$ |
| asterix | $630663.91 \pm 82753.46$ | $731452.01 \pm 71621.96$ | $\mathbf{815753.95 \pm 91022.04}$ |
| asteroids | $31957.42 \pm 15540.09$ | $53916.46 \pm 12925.49$ | $\mathbf{77368.55 \pm 17900.09}$ |
| atlantis | $1491384.23 \pm 5978.05$ | $1512404.85 \pm 10047.26$ | $\mathbf{1544673.15 \pm 5590.54}$ |
| bank heist | $\mathbf{13913.32 \pm 3529.15}$ | $11674.48 \pm 1694.59$ | $8565.69 \pm 4070.27$ |
| battle zone | $258533.57 \pm 22865.64$ | $\mathbf{321572.13 \pm 32083.18}$ | $206177.95 \pm 27251.07$ |
| beam rider | $16301.02 \pm 1853.73$ | $17465.26 \pm 4954.96$ | $\mathbf{21680.37 \pm 5991.66}$ |
| berzerk | $8359.80 \pm 201.10$ | $15824.26 \pm 5556.26$ | $\mathbf{16161.60 \pm 2848.82}$ |
| bowling | $174.27 \pm 0.10$ | $\mathbf{229.04 \pm 6.36}$ | $201.86 \pm 25.21$ |
| boxing | $\mathbf{100.00 \pm 0.00}$ | $99.99 \pm 0.01$ | $99.83 \pm 0.12$ |
| breakout | $441.21 \pm 15.08$ | $469.52 \pm 31.70$ | $\mathbf{474.30 \pm 37.00}$ |
| centipede | $178635.17 \pm 17227.15$ | $362169.27 \pm 43577.84$ | $\mathbf{525652.45 \pm 19649.69}$ |
| chopper command | $573055.88 \pm 75343.57$ | $766193.62 \pm 109233.61$ | $\mathbf{860939.78 \pm 116076.87}$ |
| crazy climber | $226821.26 \pm 3608.19$ | $224084.35 \pm 7322.03$ | $\mathbf{256189.16 \pm 21605.50}$ |
| defender | $\mathbf{540124.74 \pm 4488.40}$ | $525077.01 \pm 6084.08$ | $503190.45 \pm 32182.12$ |
| demon attack | $\mathbf{143762.91 \pm 106.75}$ | $143537.16 \pm 139.44$ | $143554.33 \pm 63.61$ |
| double dunk | $23.85 \pm 0.15$ | $\mathbf{23.90 \pm 0.07}$ | $23.81 \pm 0.11$ |
| enduro | $\mathbf{2361.56 \pm 1.03}$ | $2352.77 \pm 4.01$ | $2353.81 \pm 3.50$ |
| fishing derby | $52.58 \pm 0.32$ | $\mathbf{64.20 \pm 2.52}$ | $52.53 \pm 1.24$ |
| freeway | $\mathbf{33.79 \pm 0.08}$ | $33.69 \pm 0.13$ | $33.57 \pm 0.09$ |
| frostbite | $17692.42 \pm 2871.83$ | $\mathbf{20847.02 \pm 15492.47}$ | $19716.28 \pm 13424.59$ |
| gopher | $\mathbf{113716.78 \pm 3966.91}$ | $105370.92 \pm 12883.78$ | $101383.00 \pm 7891.20$ |
| gravitar | $\mathbf{8373.70 \pm 1260.75}$ | $8358.32 \pm 1022.94$ | $6104.39 \pm 1215.73$ |
| hero | $40825.09 \pm 3736.25$ | $\mathbf{45837.71 \pm 194.61}$ | $43076.13 \pm 4119.55$ |
| ice hockey | $60.36 \pm 4.94$ | $\mathbf{66.97 \pm 0.72}$ | $30.15 \pm 4.43$ |
| jamesbond | $1484.87 \pm 489.66$ | $1137.12 \pm 89.95$ | $\mathbf{4119.56 \pm 490.29}$ |
| kangaroo | $\mathbf{15965.79 \pm 36.61}$ | $15862.51 \pm 234.05$ | $15855.24 \pm 224.30$ |
| krull | $\mathbf{406596.00 \pm 55547.76}$ | $154118.68 \pm 179080.83$ | $350784.05 \pm 205164.12$ |
| kung fu master | $\mathbf{196638.89 \pm 456.09}$ | $193105.00 \pm 5378.00$ | $195990.25 \pm 4969.60$ |
| montezuma revenge | $12086.71 \pm 1217.76$ | $\mathbf{12714.19 \pm 824.60}$ | $11472.65 \pm 629.87$ |
| ms pacman | $10996.90 \pm 262.74$ | $\mathbf{11337.68 \pm 1176.17}$ | $10770.08 \pm 1005.72$ |
| name this game | $30252.11 \pm 884.84$ | $\mathbf{30656.34 \pm 373.97}$ | $28103.24 \pm 2023.96$ |
| phoenix | $553429.34 \pm 24278.55$ | $510548.66 \pm 236677.91$ | $\mathbf{805269.57 \pm 37169.21}$ |
| pitfall | $-0.39 \pm 0.39$ | $-0.44 \pm 0.32$ | $\mathbf{-0.01 \pm 0.02}$ |
| pong | $20.90 \pm 0.01$ | $\mathbf{20.93 \pm 0.02}$ | $20.19 \pm 0.93$ |
| private eye | $40435.54 \pm 51.04$ | $\mathbf{40472.03 \pm 39.10}$ | $40448.67 \pm 40.02$ |
| qbert | $16057.31 \pm 318.87$ | $15983.72 \pm 888.74$ | $\mathbf{17954.73 \pm 302.05}$ |
| riverraid | $28550.32 \pm 2298.03$ | $\mathbf{34591.91 \pm 831.68}$ | $34268.13 \pm 149.84$ |
| road runner | $251261.09 \pm 31741.38$ | $307342.61 \pm 41017.79$ | $\mathbf{308258.62 \pm 84372.17}$ |
| robotank | $98.45 \pm 2.85$ | $\mathbf{103.82 \pm 2.98}$ | $90.17 \pm 8.09$ |
| seaquest | $86605.86 \pm 55065.85$ | $\mathbf{259408.14 \pm 144362.40}$ | $88376.19 \pm 105086.08$ |
| skiing | $-30121.95 \pm 70.62$ | $\mathbf{-29786.38 \pm 401.06}$ | $-29878.47 \pm 289.38$ |
| solaris | $\mathbf{24366.59 \pm 4868.05}$ | $24111.78 \pm 2745.89$ | $19355.27 \pm 4102.09$ |
| space invaders | $30609.21 \pm 7141.11$ | $43675.67 \pm 6763.52$ | $\mathbf{51318.54 \pm 4277.08}$ |
| star gunner | $\mathbf{171294.31 \pm 23185.79}$ | $138390.23 \pm 6320.24$ | $135606.16 \pm 8098.51$ |
| surround | $5.86 \pm 1.44$ | $\mathbf{6.89 \pm 1.03}$ | $-5.48 \pm 0.88$ |
| tennis | $23.96 \pm 0.01$ | $\mathbf{23.97 \pm 0.01}$ | $23.86 \pm 0.06$ |
| time pilot | $44936.87 \pm 137.49$ | $\mathbf{65721.81 \pm 3213.79}$ | $53053.12 \pm 4275.77$ |
| tutankham | $\mathbf{420.36 \pm 30.13}$ | $385.05 \pm 5.02$ | $342.17 \pm 4.79$ |
| up n down | $562739.02 \pm 8527.59$ | $581518.52 \pm 5198.05$ | $\mathbf{582923.55 \pm 2849.01}$ |
| venture | $2110.64 \pm 55.39$ | $\mathbf{2308.26 \pm 14.97}$ | $2054.24 \pm 65.41$ |
| video pinball | $463141.28 \pm 426927.92$ | $133315.36 \pm 70576.86$ | $\mathbf{640269.38 \pm 330619.65}$ |
| wizard of wor | $30453.12 \pm 2470.20$ | $\mathbf{34648.51 \pm 4182.32}$ | $26076.64 \pm 2060.62$ |
| yars revenge | $280333.48 \pm 69704.31$ | $\mathbf{320777.97 \pm 64750.83}$ | $267861.48 \pm 81193.44$ |
| zaxxon | $67611.78 \pm 6226.04$ | $75165.80 \pm 4030.42$ | $\mathbf{82868.90 \pm 10160.99}$ |

Table 9: Final scores per game in our ablation study after 5B frames. We consider versions of $BT(\pi_{NGU})$ where the pre-trained policy is used for temporally-extended exploitation (flights), as an extra action (action), or both.

| Game | R2D2 | R2D2 + BT($\pi_{NGU}$) (flights) | R2D2 + BT($\pi_{NGU}$) (action) | R2D2 + BT($\pi_{NGU}$) |
|---|---|---|---|---|
| asterix | $472812.21 \pm 222663.81$ | $\mathbf{630663.91 \pm 82753.46}$ | $512869.97 \pm 109039.77$ | $618352.67 \pm 103940.46$ |
| bank heist | $965.63 \pm 133.72$ | $\mathbf{13913.32 \pm 3529.15}$ | $11052.82 \pm 6848.39$ | $12424.39 \pm 1443.95$ |
| frostbite | $9287.24 \pm 167.11$ | $9114.77 \pm 511.31$ | $10506.15 \pm 3653.44$ | $\mathbf{17692.42 \pm 2871.83}$ |
| gravitar | $6123.08 \pm 103.19$ | $6308.62 \pm 78.84$ | $7228.15 \pm 1842.07$ | $\mathbf{8373.70 \pm 1260.75}$ |
| jamesbond | $\mathbf{6056.14 \pm 1643.52}$ | $1615.21 \pm 602.84$ | $3962.10 \pm 798.68$ | $1484.87 \pm 489.66$ |
| montezuma revenge | $1478.38 \pm 1114.20$ | $11152.10 \pm 664.82$ | $6433.33 \pm 372.68$ | $\mathbf{13265.91 \pm 372.02}$ |
| ms pacman | $\mathbf{11212.85 \pm 103.23}$ | $10996.90 \pm 262.74$ | $10648.85 \pm 796.10$ | $10839.60 \pm 445.00$ |
| pong | $20.93 \pm 0.01$ | $20.90 \pm 0.01$ | $\mathbf{20.94 \pm 0.04}$ | $20.88 \pm 0.02$ |
| private eye | $23592.22 \pm 11876.55$ | $\mathbf{40492.85 \pm 27.35}$ | $37029.10 \pm 2437.54$ | $40435.54 \pm 51.04$ |
| space invaders | $3621.76 \pm 5.81$ | $\mathbf{30671.58 \pm 4418.89}$ | $3597.41 \pm 21.06$ | $30609.21 \pm 7141.11$ |
| tennis | $7.99 \pm 22.56$ | $8.00 \pm 22.54$ | $-7.15 \pm 22.00$ | $\mathbf{23.96 \pm 0.01}$ |
| up n down | $529363.05 \pm 16813.20$ | $562739.02 \pm 8527.59$ | $550665.39 \pm 1852.05$ | $\mathbf{566938.61 \pm 2428.52}$ |

Table 10: Final scores per task in Atari games with modified reward functions. We report training results for the standard game reward, a variant with sparse rewards (easy), and a task with deceptive rewards (hard). Despite the pre-trained policy might obtain low or even negative scores in some of the tasks, committing to its exploratory behavior eventually lets the agent discover strategies that lead to high returns.

| Game | R2D2 | R2D2 + $\epsilon z$-greedy | Fine-tuning $\pi_{NGU}$ | $\pi_{NGU}$ | R2D2 + BT($\pi_{NGU}$) |
|---|---|---|---|---|---|
| Ms Pacman: original | $\mathbf{11407 \pm 122}$ | $8099 \pm 868$ | $8359 \pm 2117$ | $1360$ | $10984 \pm 665$ |
| Ms Pacman: ghosts (easy) | $8375 \pm 577$ | $4322 \pm 932$ | $8356 \pm 551$ | $146$ | $\mathbf{8789 \pm 651}$ |
| Ms Pacman: ghosts (hard) | $2836 \pm 26$ | $4018 \pm 1025$ | $1891 \pm 1342$ | $-898$ | $\mathbf{7868 \pm 1085}$ |
| Hero: original | $\mathbf{43762 \pm 4918}$ | $39018 \pm 3262$ | $46848 \pm 1199$ | $9298$ | $42675 \pm 3905$ |
| Hero: miners (easy) | $3000 \pm 0$ | $3000 \pm 0$ | $3000 \pm 0$ | $1351$ | $\mathbf{4665 \pm 470}$ |
| Hero: miners (hard) | $2677 \pm 23$ | $2155 \pm 95$ | $700 \pm 0$ | $-1473$ | $\mathbf{3547 \pm 122}$ |

# I Learning curves

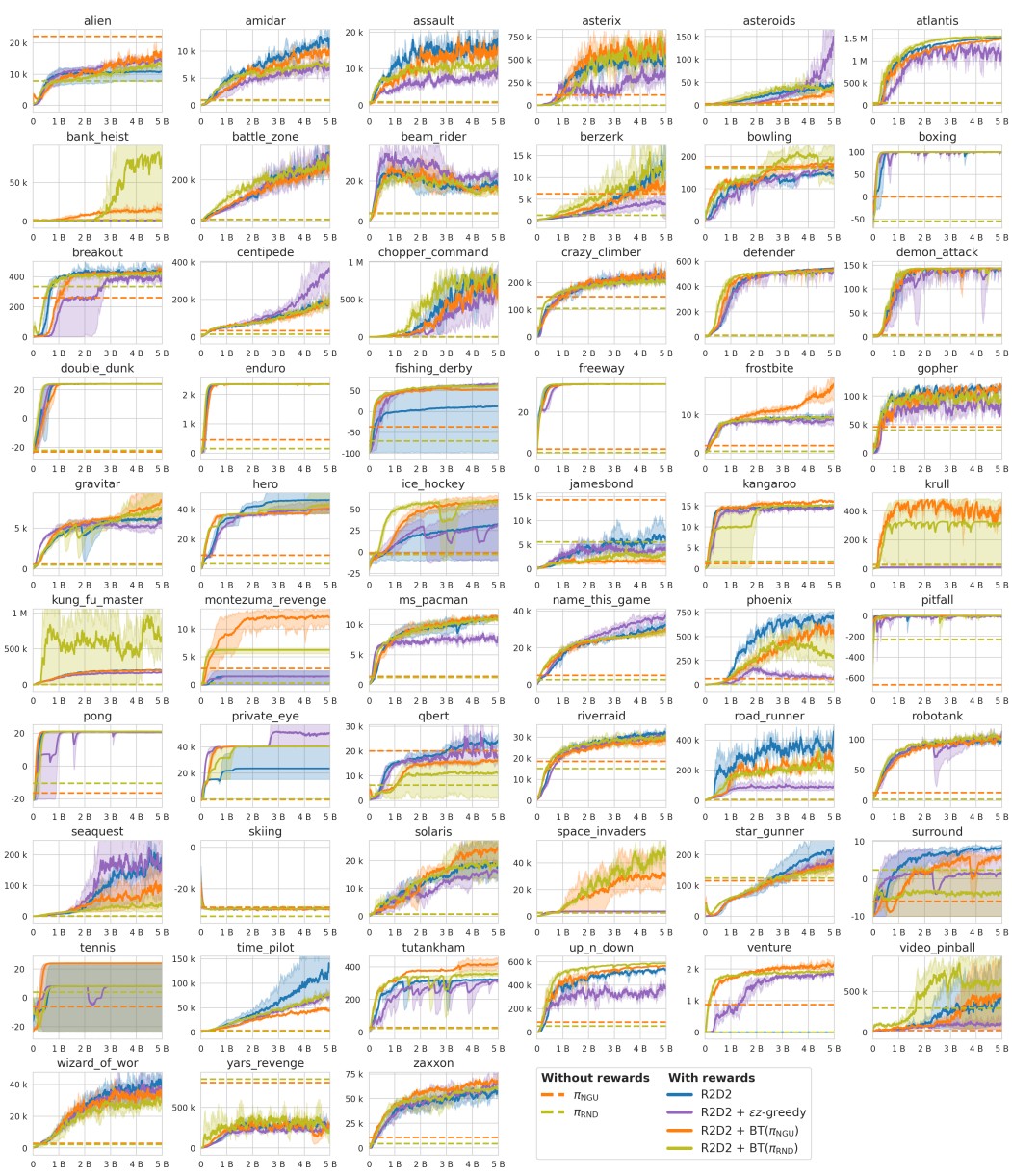

Figure 9: Training curves in all 57 Atari games for R2D2-based agents. Shading shows maximum and minimum over 3 runs, while dark lines indicate the mean.

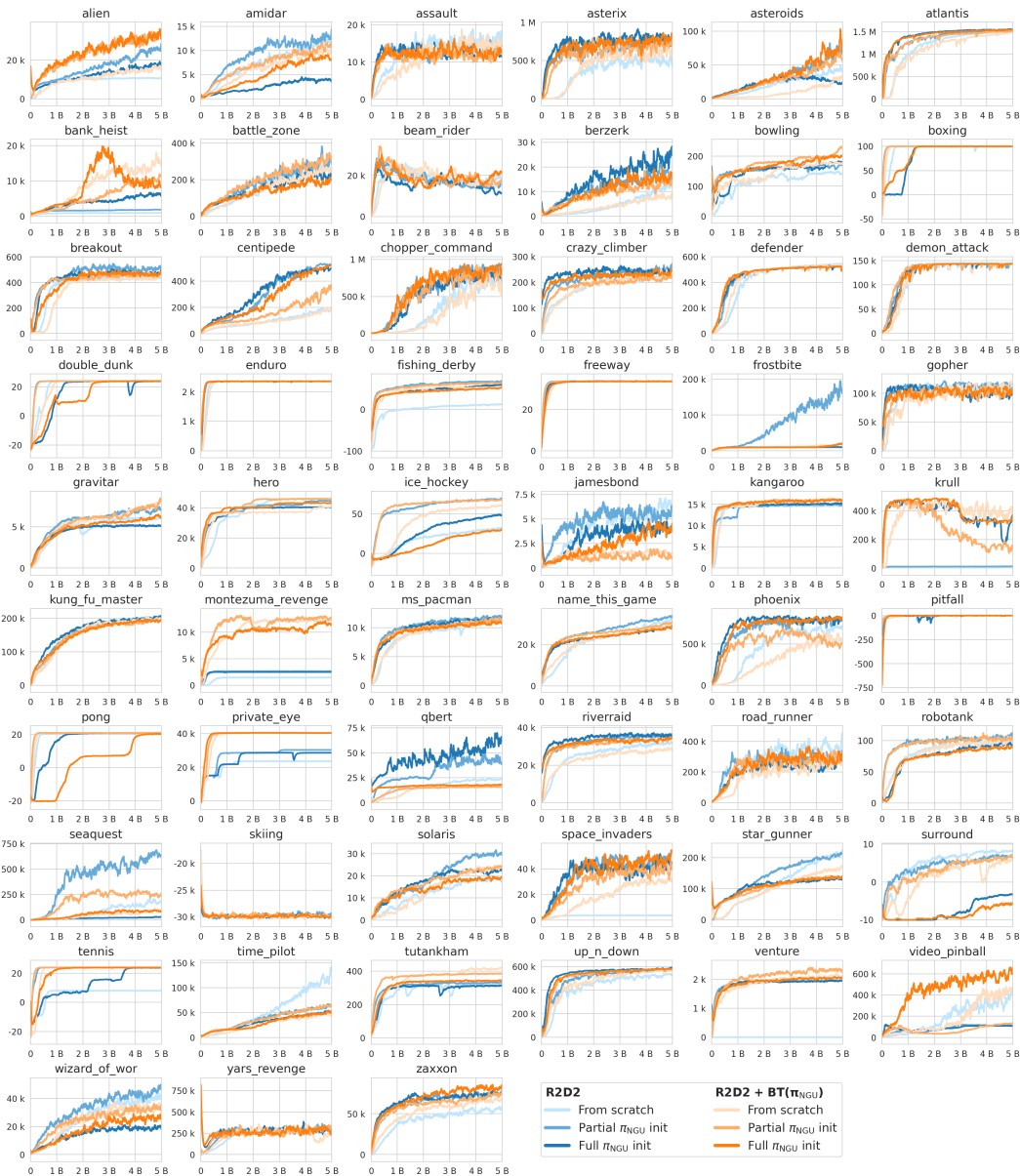

Figure 10: Training curves in all 57 Atari games for R2D2-based agents with different amounts of transfer via weights. Policies are composed of a CNN encoder followed by an LSTM and a dueling head. We compare training from scratch, loading all weights (Full $\pi_{\mathrm{NGU}}$ init) or all weights except those in the dueling head (Partial $\pi_{\mathrm{NGU}}$ init). Shading with maximum and minimum over runs is not shown for clarity, but all plots report the mean over 3 seeds.

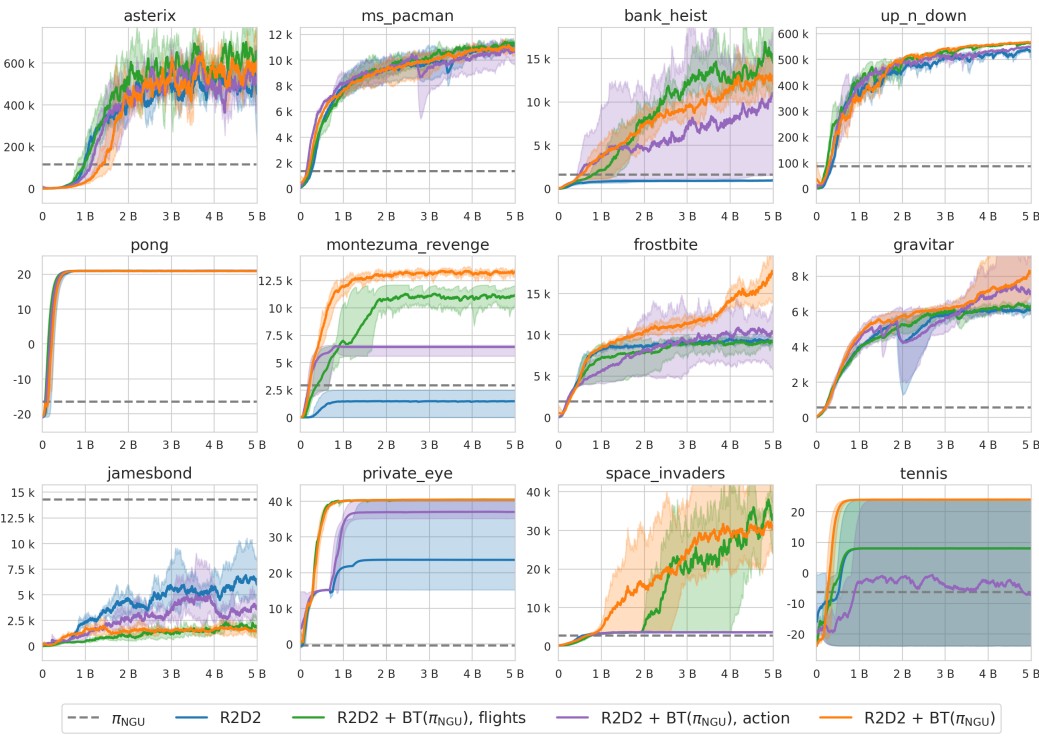

Figure 11: Training curves for ablation experiments. Shading shows maximum and minimum over 3 runs, while dark lines indicate the mean. Both ablations of BT offer benefits over the baseline, but in different sets of games. Combining them retains the best of both methods, and boosts performance even further in some games.