# OpenReview forum: "Beyond Fine-Tuning: Transferring Behavior in Reinforcement Learning"
_NeurIPS.cc/2021/Conference — NeurIPS 2021 Submitted_

### Official Review · Reviewer_sTTP · 2021-07-01

**Rating:** 7
**Confidence:** 3

**Summary:**

The authors present a simple, intuitive method for fine-tuning “behaviors” by pretraining an exploration policy before using it as a temporally extended exploration procedure and a one-step additional action to the downstream policy on a downstream task. They extensively evaluate this on the suite of Atari games and demonstrate strong performance gains. Despite some qualms I have with assumptions and ablations, I think this paper should be accepted to NeurIPS. However, I do hope that the authors address some of my concerns in the rebuttal period. For now, this is a "weak accept" but if my concerns are addressed properly the score will be raised appropriately.


**Limitations And Societal Impact:**

Yes

**Main Review:**

## Paper Strengths
**Method and Framing**
The method is simple and intuitive, and demonstrates strong performance improvements. Furthermore, the paper questions the traditional train-then-finetune paradigm that is typical in transfer learning for reinforcement learning. This is an interesting point to focus on and is useful to think about more for transfer learning in reinforcement learning in general, irrespective of the exact method.

**Clarity**
This paper is well written, with a good flow that integrates background knowledge, some related works, and the method itself within the first four sections.

**Experiments**
Demonstrating that hard exploration games like Montezuma’s Revenge benefit from more unsupervised environment frames after transfer despite it not having an effect on 0-shot transfer performance is sensible. Furthermore, Figure 4, which highlights the extra action selection distribution over training during behavior transfer, is intriguing and demonstrates that the policy really learns how to take advantage of the pre-trained exploration policies.

Overall, the number and scope of the experiments is impressive, highlighting many strengths of the method under different conditions.


## Paper Weaknesses
**Pretraining Assumption**
“Interactions between the agent and the environment are often assumed to incur a cost, but we will consider this cost to be relevant only for transitions with reward” - I’m not sure this is a valid assumption. I think the authors should include a comparison/figure which includes the total number of samples including pre-training samples for BT, as sample efficiency is important partially for reasons related to total algorithm running time. Here, having these large numbers of pre-training steps still greatly increases the algorithm running time.

**Technical Contributions**
Nothing here is particularly surprising. However, that is fine given that this method works well and is relatively easy to implement.

**Experiments**
The “Transfer to Multiple Tasks” experiment would’ve been more enlightening if there were more tasks and/or more reward functions tested. The authors clearly have the computational resources to run comprehensive experiments on the Atari suite, so it seems strange to only perform this experiment on two tasks with just two reward functions each. Can the authors experiment on more environments?

**Ablations**
Why one-step action execution of the pretrained policy? A temporally extended action should still be useful here. Can the authors should include an ablation with different length of action executions of this pretrained policy?


-- UPDATE: score revised from 6 to 7 --

**Time Spent Reviewing:**

4

---

> ### Author Response · Authors · 2021-08-10
> **Response to Reviewer sTTP**
>
> We thank the reviewer for the feedback and comments.
>
> >Pretraining Assumption “Interactions between the agent and the environment are often assumed to incur a cost, but we will consider this cost to be relevant only for transitions with reward” - I’m not sure this is a valid assumption.
>
> We want to point out that there are many settings in which the assumption holds true. For instance, agents controlling a digital environment (e.g. to control a computer) or agents trained in simulation. We believe that the transfer setting studied in this work will become more relevant as we move towards more complex environments, where we may want to train agents to maximise multiple reward functions under constant dynamics. These settings resemble the ones encountered in semi-supervised learning in Computer Vision or NLP, where rapid adaptation to new tasks is required.
>
> Several previous works have proposed methods under this assumption following [28] (reference from the manuscript). However, most proposed works do not scale gracefully with experience and computation. We believe that our contribution will bring this to attention to the community and help better connect the recent advances in semi-supervised learning with the work in transfer in deep RL.
>
> > I think the authors should include a comparison/figure which includes the total number of samples including pre-training samples for BT, as sample efficiency is important partially for reasons related to total algorithm running time. Here, having these large numbers of pre-training steps still greatly increases the algorithm running time.
>
> In our view, which to the best of our knowledge is shared by all works in this area, is that the most meaningful comparison is adaptation speed among methods that use pretraining. Which is the most consistent evaluation with our hypothesis.  Baselines training from scratch are important to put into context the speed-up gains provided by the pre-training phase. Naturally, when adapting on a single task, the overall advantage of pretraining is reduced. As the reviewer points out, in that setting the cost of the pre-training becomes significant.
>
> Our method (and works in this area) focus on task agnostic pre-training that can generalise to multiple tasks. In this case, the pre-training can be amortised. Having said that, we agree that it is also informative to include these graphs. In fact, Figure 2 attempts to show in a single graph the interaction between pretraining and adaptation. Our goal was to show the impact of pretraining but it also surface the cost of pretraining. In the updated version of the manuscript we will incorporate new graphs including the pre-training and a discussion on how the cost/benefit of pretraining changes with the number of adaptation tasks.
>
> > Technical Contributions. Nothing here is particularly surprising. However, that is fine given that this method works well and is relatively easy to implement.
>
> We thank the reviewer for this assessment. Our aim was to propose a method with complexity comparable to that of finetuning (the mechanism used by the majority of the works in the area).
>
> > Experiments The “Transfer to Multiple Tasks” experiment would’ve been more enlightening if there were more tasks and/or more reward functions tested. The authors clearly have the computational resources to run comprehensive experiments on the Atari suite, so it seems strange to only perform this experiment on two tasks with just two reward functions each. Can the authors experiment on more environments?
>
> The main bottleneck here is not computational power, but rather domain knowledge and manually coding the alternative rewards. Our goal was to create alternative reward functions that maintain the externally defined semantics of the game. Creating meaningful variants requires understanding of the environment itself, its reward function and even its source code. Thus we limited our work to building alternative reward games for two examples as proof of concept.
>
> We agree with the reviewer that the paper would be stronger if we included more variants. We believe however that the current results provide enough evidence to show that the method can properly handle many tasks and is robust to the task at hand not being aligned with the pretrained policy. Having said that we commit to incorporating alternative reward functions to more games in the revised version of the manuscript.
>
> > *Ablations* Why one-step action execution of the pretrained policy? A temporally extended action should still be useful here. Can the authors should include an ablation with different length of action executions of this pretrained policy?
>
> We agree with the reviewer that it would be informative to include an ablation in which the extra action is used in a temporally extended way. In fact, early in the project we tested this idea but we did not see benefits.
>
> We speculate that the main reasons are: $(i)$ looking at Figure 4 in the manuscript we see that early in training (when it is most useful), the agent uses the extra action a large percentage of the time. This means that in practice the use becomes temporally extended, without having to commit to a fixed length a priori (and even allowing it to change throughout the adaptation phase) $(ii)$ adding temporal extension adds another hyper parameter to tune (its length) and raises practical questions on how to estimate the value of the extra action in terms of credit assignment and discounting.
>
> In the limited amount of time that we have to provide a response, we were unable to run a baseline that implements these flights in a fair way. Having said that, this would certainly be an extension worth testing. We will add a detailed discussion on this regard in the updated version of the manuscript.

---

> > ### Comment · Reviewer_sTTP · 2021-08-16
> > **Response to authors**
> >
> > Thanks for the detailed reply. In response to your reply, and the other reviewers' reviews and your replies, I have raised my score from a 6 to a 7. I believe that with the extra experiments and proposed extended discussions proposed to be put into the paper, this paper is a solid accept.

---

### Official Review · Reviewer_fBw3 · 2021-07-12

**Rating:** 5
**Confidence:** 4

**Summary:**

This paper proposes an approach to transferring pretrained models in a deep RL context. Instead of pretraining a neural network and then simply copying the weights and retraining on a new task (as would be the naive approach taken in supervised learning), the paper instead proposes transferring the pretrained policy's behaviour. This is done by modifying the epsilon-greedy exploration strategy to include the ability to use the action suggested by the pretrained policy, which can also be used as a kind of multi-step option for improved exploration.  Pretraining is conducted using the "Never Give Up" (NGU) intrinsic reward in a reward-free setting, and experiments are conducted in the full set of Atari games. Results and ablation studies indicate that such a transfer approach can drastically improve the performance of an agent on a new task.

The main contributions are a) the use of NGU as an unsupervised pretraining objective in RL; b) a method for incorporating a pretrained policy when learning a new policy; c) experimental results and ablation studies across the full suite of Atari tasks that demonstrate the improvement resulting from leveraging the pretrained model when exploring in a new task.

**Limitations And Societal Impact:**

There are perhaps two limitations to this work that could also be discussed.

1. Adding a virtual action allows an agent to learn whether or not to reuse the pretrained policy, but it is unclear how we would immediately incorporate this into a continuous-action setting, since we would then have a mixture of continuous and discrete actions (there are, of course, approaches to overcoming this issue, but it wouldn't be straight-forward)
3. The results here show the importance of an extended pretraining period. However, this may simply not be possible given the domain - how would one collect 10 years worth of data on a single robot?

**Main Review:**


There is a lot to like about this paper. I found it extremely polished, well-written and easy to read. The proposed approach is simple, but in a good way, since it can be incorporated into existing work and codebases with very little effort. The experiments are also very extensive, with results reported over all of the Atari domains. However, I have some concerns regarding the novelty, whether the benchmark environments are appropriate, and some missing related work, which I'll discuss below.

Main comments:

1. As mentioned, one of the main positives of the paper is the ease with which the method can be integrated into existing work. Despite lacking theoretical justification, the empirical results across all the Atari games clearly show that, in practice, we can achieve far better performance with this kind of pretraining and transfer approach. I found the gap between the performance of the pretrained policy and the final policy interesting, since it clearly shows that we don't need a particularly strong pretrained policy (or that it doesn't have to be well aligned with the downstream task) for everything to be effective.

2. My main concern is that I'm not sure whether the Atari suite is the correct testbed for this method. One reason why pretraining on say ImageNet in the supervised learning setting is so useful is that the models can be reused on other domains that are not necessarily ImageNet, but which operate in the image domain. In this case, though, a policy is pretrained only on the task that we're trying to solve.  So we need to pretrain a network on Montezuma's Revenge in order to play Montezuma's Revenge, but that's all we can do with it. I think the key here is the importance of amortisation - the cost of pretraining must be paid off by being able to solve downstream tasks quickly. But in the Atari suite, we can't just pretrain one policy and then use it for any Atari game - we need to pretrain for each and every Atari game, which defeats the purpose of pretraining in the first place, since no amortising is taking place. Given this, it is perhaps unsurprising that a neural network pretrained for 16bn timesteps on game A then has subsequently improved performance on game A!  Granted, there are two modified domains that attempt to address this issue, but considering just these two limits the experimental results and there's no display of amortisation.
If the aim is to stick with Atari, then a more convincing approach would be to have a single pretrained model that we can use to solve any subsequent Atari game - the cost of pretraining could then be amortised over 57 subsequent tasks! However, I feel a better domain would have been something like DeepMind lab, where the "base" MDP captures the dynamics of the open world, and then subsequent downstream tasks would all be conducted in the same environment (e.g. collecting different objects). This to me would be more analogous to the ImageNet case, since we could then reuse the policy for each task within the DeepMind lab domain, much as we can reuse the ImageNet model in any natural image task. I realise that suggesting a completely different experimental setup is not immediately feasible, so I would also be open to discussion on why the current Atari experiments are appropriate.


3. Another issue I'm unsure of is the novelty and contribution of the work, which seems to be combining existing techniques. I thought the discussion of an appropriate pretraining strategy was interesting (since other approaches have attempted to do things like autoencoders etc), and I think this could open many avenues for future work regarding how best to do so in the deep RL setting. However, the actual method of behavioural transfer is a combination and modification of existing techniques, which limits the novelty. For example, using a different policy to solve a new task by modifying the exploration behaviour has been done in prior work (e.g. [1] who call it \pi-reuse). The difference here is simply that a) the policy can be used as an n-step option, and b) by adding in a virtual action, the agent can learn whether or not to explicitly use the pretrained policy. The contribution then feels like it's mainly about putting together these existing ideas and showing that they work well empirically.

4. While there's a good discussion on prior work, I was very surprised to see no mention of reward-free RL [2], which is exactly the setting considered here (agent learns in a domain with no rewards, and then must quickly learn a downstream task after this pretraining phase). Although much of the work in this space is restricted to tabular [2, 3, 4] or linear function approximation [5], the advantage with these approaches is that they have guarantees. While not directly comparable to this work, I think it is definitely worth mentioning them.

Minor comments:

1. Future work could potentially focus on other ways of incorporating the pretrained policy, such as using entropy regularised RL or similar.

2. On line 164, should it not read that you update a^+ as well as the action chosen by the pretrained policy? At the moment, you have \pi(s) and a', which are the same thing.

3. A couple of the references have arxiv versions, whereas published ones exist (e.g. Brown et al. [9] was at NeurIPS 2020)

[1] Fernández, Fernando, and Manuela Veloso. "Probabilistic policy reuse in a reinforcement learning agent." Proceedings of the fifth international joint conference on Autonomous agents and multiagent systems. 2006.
[2] Jin, Chi, et al. "Reward-free exploration for reinforcement learning." International Conference on Machine Learning. PMLR, 2020.
[3] Zhang, Zihan, Simon Du, and Xiangyang Ji. "Near Optimal Reward-Free Reinforcement Learning." International Conference on Machine Learning. PMLR, 2021.
[4] Zanette, Andrea, et al. "Provably efficient reward-agnostic navigation with linear value iteration." NeurIPS, 2020.
[5] Wang, Ruosong, et al. "On reward-free reinforcement learning with linear function approximation." NeurIPS, 2020.


***POST-REBUTTAL***

Thank you for the response, the comparison to prior work and additional experiments that showcase the importance of the *multistep* flight. Despite this, I still have concerns about the use of Atari which result in the following shortcomings (ignoring the Ms PacMan results) :

1. Due to the use of Atari, transfer here is restricted to single target task instances (source task is the game without reward and the target task is the game with reward). However, multitask transfer (for target tasks with different reward functions/dynamics) could have been shown by using a different testbed like Marion, ProcGen, Retro, etc).
2.  Related to this, the motivation in the introduction is that computer vision benefits from this pretraining. However, that's because that these models can transfer to multiple target tasks, not just one. The fact that the multitask transfer does not occur here is a limitation of  Atari, not the approach, which seems like a waste.
3. I'm not sure I can agree with the assertion that pretraining is "free" because there's no reward function. Why should that be?  Better motivation here would be helpful. But again, if we just had the ability to transfer to multiple tasks, then the pretraining costs could be recouped in the long run.

If Atari is a must, one final suggestion would be to use the different game "modes" offered by the ALE (Machado et al, 2018). Then, at the very least, multitask transfer could be shown on the different version of the same game.


**Time Spent Reviewing:**

8

---

> ### Author Response · Authors · 2021-08-10
> **Response to Reviewer fBw3**
>
> We thank the reviewer for the feedback and comments.
>
> > 2. In this case, though, a policy is pretrained only on the task that we're trying to solve. So we need to pretrain a network on Montezuma's Revenge in order to play Montezuma's Revenge, but that's all we can do with it. [...]
> > Granted, there are two modified domains that attempt to address this issue, but considering just these two limits the experimental results and there's no display of amortisation.… I realise that suggesting a completely different experimental setup is not immediately feasible, so I would also be open to discussion on why the current Atari experiments are appropriate
>
> (Part of this answer was also shared with Reviewer 1MjX)
>
> The Atari-57 suite is one of the most widely used benchmarks in deep RL. Many works exploring ideas of transfer learning in RL have used it [10, 28, 35, 41] (references from the manuscript). Naturally the use of this well-understood collection of tasks lets us position our contribution more clearly against the current literature.
>
> Apart from this, we believe that there are valid arguments behind this choice. The Atari-57 suite provides a large number of diverse games. This suite was originally proposed as a good setting for evaluating general competency in AI agents [7] (reference from the manuscript). The Atari 2600 games are varied enough to claim generality. Each game is interesting enough to be representative of settings that might be faced in practice, and (perhaps most importantly) each game was created by an independent party and is free of experimenter’s bias.
>
> We understand the reviewer’s concerns. We agree with the reviewer that the ideal setting for testing ideas of transfer learning are environments in which, given fixed dynamics, one has access to a large number of extrinsically defined tasks of interest. In such a setting, the benefits of the amortisation of the pre-training become apparent. We agree that Atari has limitations in that sense (for each game environment there is a single reward function), and this is exactly what we tried to achieve with the “Transfer to multiple tasks.” set of experiments.
>
> The reviewer states that DeepMind lab would provide a good example of an environment to test transfer learning, as many tasks could be defined. We agree that this is a strong point, but we think that it has complementary pros and cons as the Atari suite. DM lab represents a single instance of an environment that was designed to perform deep RL research. Testing on a single domain (with its particularities) might limit the generality of the results.
>
> Considering the above statements, we believe that the underlying problem highlighted by the reviewer is out of the scope of our work. The community is in need of a good test suite for unsupervised and transfer learning in RL containing a large number of diverse environments each of which with a large quantity of meaningful externally defined tasks.
>
> We will include a detailed discussion of the points raised by the reviewer in the revised version of the manuscript.
>
> > 3. […[ using a different policy to solve a new task by modifying the exploration behaviour has been done in prior work (e.g. [1] who call it \pi-reuse). The difference here is simply that a) the policy can be used as an n-step option, and b) by adding in a virtual action, the agent can learn whether or not to explicitly use the pretrained policy. The contribution then feels like it's mainly about putting together these existing ideas and showing that they work well empirically.
>
> We thank the reviewer for pointing out reference [1] (reference as cited by the reviewer). We will incorporate it into the manuscript and add a detailed discussion of the differences with the proposed approach.
>
> First we want to highlight that our work presents evidence for the importance of large scale pretraining in unsupervised RL and also highlights limitations in the way finetuning transfers behavior. Which are contributions that go beyond the proposed method, BT.
>
> Regarding the methods themselves, we acknowledge they are very related. We agree with the differences stated by the reviewer and would like to add some further comments.
>
> - The method in [1] utilises past policies that solve similar tasks within the same domain. Meaning, the work relies on having a sequence of externally defined tasks that share some common structure.
> - The fact that BT can successfully learn transferable behavior without the need of a library of past-policies trained on related tasks is in our view a strong result that deserves to be shared with the community.
> - In the ablation study presented in Section 5, we observe that removing either one of the mentioned modifications drastically affects performance. To further stress this point, we performed extra experiments removing the temporal extension in BT's exploratory flights (please find a summary of the results below). In short, while being small algorithmic changes, they are conceptually and empirically very important, e.g. exploration needs to be directed and temporally extended to be most effective.
> - The method in [1] was tested on tabular domains. From that paper it is unclear whether it would transfer to settings with function approximation in the large deep RL settings tested in this work.
> - As [1] is concerned with tabular settings, that work does not explore (or formulate) its relation to modern transfer mechanisms based on weight sharing.
> - Finally, given that most current works predominantly use fine-tuning, it would be appropriate to resurface this work to the community along with the empirical and methodological contributions of our work.
>
> We ran a variant of our algorithm in which exploratory flights are restricted to a single step. The table below summarises the findings (all variants of BT use the NGU pre-trained policy).
>
> | | Mdn@1B |  M@1B   | Mdn@5B |  M@5B
> | --- | --- | --- | --- | --- |
> *BT($\pi_{\textrm{NGU}}$) with 1 step*          |   170.0    |     457.1    |    241.0   |   1214.6
> BT($\pi_{\textrm{NGU}}$)  | **279.6**| **554.8** | **360.5** | **1528.0**
> BT($\pi_{\textrm{NGU}}$) (without flights)   |   141.9    |     545.9    |    224.4   |   1275.4
> BT($\pi_{\textrm{NGU}}$) (without action)   |   186.9    |     631.7    |    223.3   |   1524.0
>
> The table presents the results (averaged over three seeds) in terms of median and mean Human Normalised Score (HNS).We report results at 1B and 5B frames. (Mdn@1B means median HNS at 1B frames)
>
> We can see that BT with 1 step performs better than the version that does not include flights at all. However, it does not recover the performance of full BT (with flights and the extra action). Thus showing that the temporal extension is a crucial aspect of the proposed method. Our observations were in line with that reported in [14] (reference from the manuscript). We will include these results as well as learning curves.
>
> > 4. While there's a good discussion on prior work, I was very surprised to see no mention of reward-free RL [2], which is exactly the setting considered here (agent learns in a domain with no rewards, and then must quickly learn a downstream task after this pretraining phase). [...] While not directly comparable to this work, I think it is definitely worth mentioning them.
>
> We thank the reviewer for bringing this to our attention. We agree that it would be relevant to mention this line of work and discuss its relation to the ideas presented in our work. We will include these references and a detailed discussion in the revised version of the manuscript.
>
> > Adding a virtual action allows an agent to learn whether or not to reuse the pretrained policy, but it is unclear how we would immediately incorporate this into a continuous-action setting, since we would then have a mixture of continuous and discrete actions (there are, of course, approaches to overcoming this issue, but it wouldn't be straight-forward)
>
> We believe that the reviewer has raised a good point. We agree that BT is not immediately transferable to continuous action spaces. Extensions in that direction would require further investigation. We will incorporate this point in the discussion of the limitations of BT in the updated version of the manuscript.
>
> > The results here show the importance of an extended pretraining period. However, this may simply not be possible given the domain - how would one collect 10 years worth of data on a single robot?
>
> Indeed one of the hypotheses of our work is that large scale pre-training is very cheap (as discussed above). We agree that applying our method directly to a robotics problem is not directly feasible. Other environments (such as fully digital ones) are better suited. We will add this as a limitation. However, we want to emphasise that research in this direction could still become important for robotics in the future. For instance, one could think of learning the pre-trained exploratory policy in simulation and combine it with sim-2-real settings.
>
> > Minor comments:
> > Future work could potentially focus on other ways of incorporating the pretrained policy, such as using entropy regularised RL or similar.
>
> Thanks. This is indeed a good suggestion.
>
> > On line 164, should it not read that you update $a^+$ as well as the action chosen by the pretrained policy? At the moment, you have $\pi(s)$ and $a'$, which are the same thing.
>
> Not that $\pi_p(s)$ is the pre-trained policy which chooses primitive actions (from the original action set), while $a'$ refers to the "extra action". What this means is that we assign credit to both the extra action and the original action, so that the agent can eventual learn not to use $\pi_p$ any longer.
>
> > A couple of the references have arxiv versions, whereas published ones exist (e.g. Brown et al. [9] was at NeurIPS 2020)
>
> Thanks for pointing this out.

---

### Official Review · Reviewer_1MjX · 2021-07-14

**Rating:** 6
**Confidence:** 3

**Summary:**

The authors study the problem of transferring pre-trained behavior for exploration in reinforcement learning. They propose an approached called (BT) which relies on the pre-trained policy for collecting experience through temporally-extended exploration, which can be triggered with some probability at any step, or  one-step calls to the pre-trained policy based on value estimates.
The experiments presented in this work show that, when combined with large-scale pre-training in the absence of rewards, existing intrinsic motivation objectives can lead to the emergence of complex behaviors.

**Limitations And Societal Impact:**

Only one limitation of BT was mentioned in the discussion section (that BT assumes that flights can be started from any state and still produce meaningful behavior.). There are no negative societal impact as far as I'm concerned.

**Main Review:**

*Originality:* The authors have proposed a novel approach for transferring pre-trained behaviour for exploration. While they use existing unsupervised pretraining using an existing method (NGU), how pre-trained policy is applied to downstream RL tasks with reward is novel. The authors made it very clear how their work differs from previous contributions and have adequately cited relevant papers.

*Quality*: The submission is technically sound. The claims are supported using the empirical results.

*Clarity*: The manuscript is well written and well organised. My only questions related to the Ablation studies Section. Specifically, in the first few sentences of this section, the authors talk about some results without proper reference to any figures. Please clarify this and refer to any figure which illustrates this.

*Significance*: I think that the results are important and useful for the RL community. The problem setting itself is very well motivated. However, I have the following concerns that I'd like the authors to address:

- The methods are only tested on the Atari benchmark. While I understand that this is what other work in transfer learning might also use for evaluation, I believe the paper can be strengthened if the methods are tested in more interesting environments. Atari tasks are singletons (every episode look the same) and deterministic, whereas for RL methods to generalise to real-world settings, approaches need to handle generalisation and stochasticity. Therefore, I'd suggest the authors try using some procedurally generated environments, such as [OpenAI ProcGen](https://github.com/openai/procgen), [NetHack](https://github.com/facebookresearch/nle), [MiniGrid](https://github.com/maximecb/gym-minigrid) or [MiniHack](https://openreview.net/forum?id=skFwlyefkWJ). In case these approaches don't work well on procedurally-generated tasks, I'd like the authors to explain why is this the case and how this can be improved in future work.

- I'd also be interested in seeing baselines transfer learning results using [Kickstarting](https://arxiv.org/abs/1803.03835) from the pre-trained policy. Since kickstarting is also agnostic to how pre-trained policy is trained, it can be used for transfer learning.

**Time Spent Reviewing:**

4

---

> ### Author Response · Authors · 2021-08-10
> **Response to Reviewer 1MjX**
>
> We thank the reviewer for the feedback and comments.
>
> > Originality: The authors have proposed a novel approach for transferring pre-trained behaviour for exploration. While they use existing unsupervised pretraining using an existing method (NGU), how pre-trained policy is applied to downstream RL tasks with reward is novel. The authors made it very clear how their work differs from previous contributions and have adequately cited relevant papers.
>
> >Quality: The submission is technically sound. The claims are supported using the empirical results.
>
> > Clarity: The manuscript is well written and well organised.
>
> We thank the reviewer for this assessment.
>
> > My only questions related to the Ablation studies Section. Specifically, in the first few sentences of this section, the authors talk about some results without proper reference to any figures. Please clarify this and refer to any figure which illustrates this.
>
> The results presented in this section study the importance of each of the two components of BT. We restrict the analysis to a subset of 12 representative games (without transferring any weights). We compare full BT against an R2D2 baseline (for reference) and two ablated versions of BT: *(i)* a version that does not include an extra action *(ii)* a version that does not include the exploratory flights. Results per game are shown in Table 9 of Appendix H.
>
> To answer a question of Reviewer fBw3, we decided to incorporate a third ablation of BT which employs the extra action but restricts exploratory flights to have a single step (please find the results in the answer to Reviewer fBw3).
>
> We will incorporate these clarifications in the manuscript. We will also incorporate in the Appendix a table summarizing the results in terms of HNS as described in the text of that section.
>
> > The methods are only tested on the Atari benchmark. While I understand that this is what other work in transfer learning might also use for evaluation, I believe the paper can be strengthened if the methods are tested in more interesting environments.
>
> (Part of this answer was also shared with Reviewer fBw3)
>
> We agree with the reviewer, the paper would be stronger with more environments. Below we explain the decision behind our choice, and leave for future work testing on new environments as we do not have the time to set up the new experiments during the rebuttal period. We will comment on this in the discussion section.
>
> The Atari-57 suite is one of the most widely used benchmarks in deep RL. Many works exploring ideas of transfer learning in RL have used it [10, 28, 35, 41]. As the reviewer states, the use of this well-understood collection of tasks lets us position our contribution more clearly against the current literature.
>
> The Atari-57 suite provides a large number of diverse games. This suite was originally proposed as a good setting for evaluating general competency in AI agents. The Atari 2600 games are varied enough to claim generality [7] (reference from the manuscript). Each game is interesting enough to be representative of settings that might be faced in practice, and (perhaps most importantly) each game was created by an independent party and is free of experimenter’s bias.
>
> Given the above, we believe that the underlying problem highlighted by the reviewer is out of the scope of our work. The community is in need of a good test suite for unsupervised and transfer learning in RL containing a large number of diverse environments each of which with a large quantity of meaningful externally defined tasks.
>
> > Atari tasks are singletons (every episode look the same) and deterministic, whereas for RL methods to generalise to real-world settings, approaches need to handle generalisation and stochasticity.  Therefore, I'd suggest the authors try using some procedurally generated environments, such as OpenAI ProcGen, NetHack, MiniGrid or MiniHack. In case these approaches don't work well on procedurally-generated tasks, I'd like the authors to explain why is this the case and how this can be improved in future work.
>
> We agree with the reviewer that it would be quite valuable to test BT (and other transfer learning approaches) to the setting described. We want to highlight that this setting (procedurally generated environments) is a generalisation of the approach commonly studied in transfer learning, where the environment is assumed to be fixed. This would certainly put more pressure on pre-training mechanisms. Since BT mainly concerns how to transfer behavior, we believe that this question is out of the scope of our work. Having said that, find below our take on the question.
>
> We believe that our approach would work well in procedurally generated tasks. The mechanism for transfer, namely the exploratory flights and incorporating an extra action, should not be fundamentally affected by the task being procedurally generated or stochasticity in the environment. As long as the pre-trained policy is able to explore the environment and the agent can learn from off-policy data, the method should work as expected. The drawbacks observed for finetuning are also expected to stay in these settings, as they have to do with not explicitly handling how the pre-trained behavior is used in the transfer phase (“overwriting” the pre-trained behavior).
>
> On the other hand, these aspects could indeed affect the quality of the pre-trained policy. The episodic nature of NGU makes it very well suited to handle procedurally generated tasks, as seen in the example shown in Section 4.1 of that paper [48] (reference from the original manuscript). RND on the other hand, seems less prepared for these scenarios as it estimates life-long state visitations from observations.
>
> We will include a detailed discussion of this issue, and incorporate as one of the limitations that BT relies on having a policy that can express complex behavior. We will further comment on the future direction of moving transfer to handle a distribution of environments with fixed dynamics.
>
> > I'd also be interested in seeing baselines transfer learning results using Kickstarting from the pre-trained policy. Since kickstarting is also agnostic to how pre-trained policy is trained, it can be used for transfer learning.
>
> Kickstarting has been proposed as a mechanism to transfer knowledge between agents solving the same task (with extrinsic rewards). For instance to train a multi-task agent using “teachers” trained on single-task, or to use a faster to train agent to help kickstart a slower one.
>
> As the reviewer mentioned, the transferring mechanism in BT is agnostic to how the policy was pre-trained. However it is unclear to us the effectiveness of Kickstarting when transferring between two different tasks. In the adaptation phase, the policy will be regressed to resemble the pre-trained task-agnostic policy. It is unclear how detrimental this would be when there is a misalignment between the pre-trained policy and good policies for the task at hand. Another aspect that favours BT over kickstarting is that it is easy to imagine an extension of BT to handle multiple pre-trained policies, whereas this is not as clear for kickstarting.
>
> We will include a detailed discussion on this topic. We are unable to provide results now as it would be very difficult for us to properly implement (and tune) a kickstarting baseline in the limited amount of time we have to prepare our response. This is further complicated by the fact that kickstarting is a process designed for policy based methods while our work (and codebase) concentrates on value-based ones.

---

> > ### Comment · Reviewer_1MjX · 2021-08-16
> > **Reply to the authors**
> >
> > I thank the reviewers for their detailed response.
> >
> > Regarding the choice of the environment, I "appreciate" the authors explaining to me how "great" Atari-57 is. And why did the authors break my point into two parts and response to them individually, rendering them out-of-context. I said, very rightfully, that Atari benchmark only proposes a limited complexity due to them being singleton and deterministic, whereas procedurally generated environments are more suited for evaluating the generalisation of agents to unseen situations, something that is needed for real-world use-cases of RL.
> >
> > Despite authors believing that their approach would work well in procedurally generated tasks, I'm afraid I would need to see some empirical evidence of this to be convinced.
> >
> > I will therefor keep my score as is at this stage.

---

> > > ### Author Response · Authors · 2021-08-19
> > > **Response to the reviewer**
> > >
> > > We apologise for the misunderstanding. We split the question to first address why we decided to use Atari as our test environment, and then to specifically talk about the use alternative environments. By no means we intended to take the raised points out of context.
> > >
> > > We start by emphasising that we agreed with the reviewer that the community is in need of a good test suite for unsupervised and transfer learning in RL. We acknowledge that Atari-57 falls short in many ways. However, we believe that it is still a valuable tool and that it is enough to back the claims made in our work. It is also what allows comparing the proposed method to prior work.
> > >
> > > In our view, algorithms should be tested in several complex environments with different types of downstream tasks. As the reviewer points out, the complexity of the environment would allow testing the generalisation of the agent to unseen (but consistent) situations. The reviewer is indeed correct that each Atari game is a singleton environment and almost deterministic (the random no-ops at the beginning makes the starting state random).
> > >
> > > Our goal in the response was to highlight that the Atari suite has 57 singleton environments with quite different dynamics. The tasks themselves are very diverse covering many different aspects of the RL problem (e.g. credit assignment or exploration). This allows testing the versatility of the adaptation mechanism. In short while each environment might be less general than procedurally generated environments, there is significant value in having a large set of environments with diverse dynamics. We will add the limitations raised by the reviewer to Section 6.

---

### Official Review · Reviewer_FMRN · 2021-07-15

**Rating:** 6
**Confidence:** 4

**Summary:**

This paper studies the transfer of unsupervised RL agents in the Atari-57 benchmark suite.

The authors propose behavior transfer (BT), a simple method for transfer that focuses on using the behavior of the pre-trained policy, separately from using the pre-trained policy as an initialization (a.k.a. fine-tuning). BT augments a downstream off-policy learner in two ways: the downstream learner can use the pre-trained policy to explore (for a temporally extended duration), and the pre-trained policy is also added as a pseudo-action to the action space as an option during exploitation (for a single time step).

In the experiments, the authors primarily focus on using the never give up (NGU) objective to obtain the pre-trained policy and assess the impact of BT with this pre-trained policy on downstream learning via recurrent replay distributed DQN (R2D2). The protocol differs from prior works in that each learning phase is much longer: 16B frames per game for pre-training (a 64x increase over [28]) and up to 5B frames per game for transfer (a 12500x increase over [35]). This is shown to be important overall. For standard Atari-57, transfer via BT compares favorably against R2D2 baselines that lack unsupervised pre-training, with more pronounced gains for hard exploration games. Ablations show that both BT methods of using the pre-trained policy are necessary for best performance. The authors also assess on custom tasks in the Ms Pacman and Hero games, with BT causing significant improvement over R2D2 baselines despite the zero-shot pre-trained policy used for BT not achieving any success. Finally, the authors also assess BT with fine-tuning from (some subset of) the pre-trained policy's weights. For standard Atari-57, fine-tuning improves over training from scratch for both R2D2 as well as R2D2+BT. However, finetuning without BT from a partial initialization outperforms any variant involving BT, an interesting negative result.

**Limitations And Societal Impact:**

Limitations are discussed above. The potential societal impact of the work was not discussed by the authors.

**Main Review:**

## Originality
The proposed method, behavior transfer (BT), is, to my knowledge, new. However, it may be helpful to reference and discuss prior works such as [A], which also broadly consider the use of pre-trained behavior in adaptation to new tasks, albeit in a different domain (simulated robotics) and with different problem assumptions (e.g. acquisition of pre-trained behavior via demonstrations). While Atari-57 is a well-established RL benchmark, the authors do bring a new perspective to unsupervised pre-training for Atari-57 by demonstrating the effect of scale in both the unsupervised pre-training and supervised adaptation phases. The authors also generate new reward functions to assess transfer to tasks other than the original games, though this is only done for two games.

## Quality
Given that the novelty of BT is the use of pre-trained policies in specific ways during subsequent off-policy RL and not the intrinsic motivation objectives used to train the policies, I was surprised that the R2D2 baseline agents were run without RND/NGU. To me, this seems vital in order to observe the effect of BT in a controlled manner so that we know what to attribute any improvements to. Otherwise, the experiments appear to be thoroughly executed and documented. There are no theoretical claims to assess.

## Clarity
The writing and presentation of the method and results are clear. However, I am not sure that the contributed method is appropriately named. Certainly, fine-tuning a pre-trained policy involves "behavior transfer" since the behavior of the pre-trained policy collects the data used for transfer learning, but fine-tuning, as the authors point out, is orthogonal to the proposed method. BT also seems too broad for any specific method. I would further recommend updating the title and/or abstract to reflect the scope of the method and experiments, specifically the focus on off-policy RL and Atari-57.

## Significance
Given the simplicity of BT, I imagine that future works will use or build on it. Getting transfer to work well in the context of RL is an open question, and the idea that transferring a pre-trained policy can involve mechanisms distinct from fine-tuning is an interesting idea that I expect the field to consider going forward. However, the fact that BT with finetuning is not the best variant for the overall Atari-57 suite should give us pause. The direct impact of this work is also severely limited by the execution and scope of its experiments: aside from not controlling for the use of intrinsic motivation/exploration objectives, it's not clear whether the effective use of pre-trained policies for exploration is specific to pre-training objectives that focus on encouraging exploration.

## References
[A] Singh et al., Parrot: Data-Driven Behavioral Priors for Reinforcement Learning, ICLR 2021.


**Time Spent Reviewing:**

10 hours

---

> ### Author Response · Authors · 2021-08-10
> **Response to Reviewer FMRN**
>
> We thank the reviewer for the feedback and comments.
>
> > The proposed method, behavior transfer (BT), is, to my knowledge, new. However, it may be helpful to reference and discuss prior works such as [A].
>
> We agree with the reviewer, reference [A] (reference from the review) is relevant prior work. While their approach uses a diverse multi-task dataset, they too aim at learning a behavioral prior to accelerate acquisition of new skills. We thank the reviewer for pointing it out. We will add the reference and a discussion in Section 6.
>
> >  Given that the novelty of BT is the use of pre-trained policies in specific ways during subsequent off-policy RL and not the intrinsic motivation objectives used to train the policies, I was surprised that the R2D2 baseline agents were run without RND/NGU.
>
> We thank the reviewer for this suggestion. We ran an NGU baseline on all 57 games. We observe that NGU trained from scratch is significantly slower than BT.
>
> The original work in NGU trains a family of different policies with different degrees of exploratory behavior. Each policy has specific hyper-parameters, namely discount factor and reward weight, to allow the agent to cope with many different environments. We ran an NGU agent with 1 and 32 policies, as the single policy setting is the one that is closest to our setting (BT could also benefit from training multiple policies, but it is left for future work).
>
> The tables below present the results in terms of median and mean Human Normalised Score (HNS). For convenience we only list two variants of BT. We report average results over three seeds at 500M, 1B and 5B frames. (Mdn@0$.$5B means median HNS at 500M frames). Results:
>
> **Full Atari-57:**
>
> | | Mdn@0$.$5B	 |  M@0$.$5B  | Mdn@1B |  M@1B   | Mdn@5B |  M@5B
> | --- | --- | --- | --- | --- | --- | --- |
> NGU (1)                         |        29.2      |       88.4    |      60.0    |     305.0    |    155.3   |   1286.6
> NGU (32)                        |       56.7      |     245.6    |    110.3    |     571.1    |     541.9   |  1919.5
> BT($\pi_{\textrm{NGU}}$) |     139.7      |     955.9    |    273.5    |    1517.1   |     566.2  |   **2261.7**
> BT($\pi_{\textrm{NGU}}$) partial init |  **274. 4** | **1664.2** |  **345.7**|   **2019.9**   |  **626.3** |  1966.8
>
>
> **Hard exploration games:**
>
> | | Mdn@0$.$5B	 |  M@0$.$5B  | Mdn@1B |  M@1B   | Mdn@5B |  M@5B
> | --- | --- | --- | --- | --- | --- | --- |
> NGU (1)                         |          1.5      |        0.8     |       9.0     |         8.7    |      51.6    |    49.1
> NGU (32)                        |         6.3      |      12.0     |     27.0     |       30.9    |      75.6    |    75.9
> BT($\pi_{\textrm{NGU}}$) |       67.0      |      68.0     |    100.8    |       94.2    |     192.0   |   159.5
> BT($\pi_{\textrm{NGU}}$) partial init | **87.2** | **102.9** | **115.4** |  **120.6** |  **200.3** |  **164.5**
>
> BT provides large gains at the beginning of the adaptation and it is particularly so in hard exploration games, even when compared to the 32-policy setting. We will also incorporate learning curves for this baseline in the updated version of the manuscript. Specifically we will include it to Figures 3, 5, and 6.
>
> Finally, we want to highlight that the exploration performed by NGU is task dependent (as the “explorer policies” optimise an augmented reward $r_e + \lambda r_i$). Nevertheless, BT is able to obtain large gains with a task agnostic exploration mechanism.
>
> > However, I am not sure that the contributed method is appropriately named. Certainly, fine-tuning a pre-trained policy involves "behavior transfer" since the behavior of the pre-trained policy collects the data used for transfer learning, but fine-tuning, as the authors point out, is orthogonal to the proposed method. BT also seems too broad for any specific method. I would further recommend updating the title and/or abstract to reflect the scope of the method and experiments, specifically the focus on off-policy RL and Atari-57.
>
> We agree with the reviewer that BT requires the underlying RL learning algorithm to handle off-policy data. We will change the abstract to highlight this important aspect of the method. We will also update the abstract to incorporate details of the scope of the experimental evaluation (Atari-57).
>
> We selected this title in the hope of highlighting the importance of explicitly handling the transfer of behavior in the RL setting. This is, to highlight that in finetuning (the mechanism used by the large majority of works in the subject) both aspects are combined in a rather uncontrolled way. Then we show that a very simple method (BT) explicitly handling the transfer of behavior can lead to better results in terms of the speed of adaptation to an unknown task. Our hope is that this will motivate researchers working in the area to look at better ways of transferring behavior, and consider the particular solution that we proposed as a first step in that direction.
>
> > Given the simplicity of BT, I imagine that future works will use or build on it. Getting transfer to work well in the context of RL is an open question, and the idea that transferring a pre-trained policy can involve mechanisms distinct from fine-tuning is an interesting idea that I expect the field to consider going forward.
>
> We thank the reviewer for this comment.
>
> > However, the fact that BT with finetuning is not the best variant for the overall Atari-57 suite should give us pause.
>
> We respectfully disagree with the take away that BT with finetuning is worse than the baseline in the Atari-57 suite.
>
> It is certainly true that BT with finetuning achieves lower scores in terms of Median HNS across the Atari-57 suite. However this is a single statistic computed over the performance of the algorithms across many games, most of which have dense rewards. It has been recently shown in [47] (reference from the manuscript) that median or mean HNS over the whole Atari suite can be deceiving: while some algorithms seem to perform really strongly when considering these statistics, they do so by achieving outstanding scores in dense reward games and failing completely on hard exploration tasks.
>
> If we look at the performance of BT+finetuning compared to the baselines, we see that it significantly improves performance on the harder settings without significantly hurting performance on dense reward games. Tables 4 in Appendix D shows that the performance of BT+finetunning across the whole suite is comparable with that of the best method (in fact, BT+finetunning achieves the highest performance in terms of mean HNS) while the BT variants clearly outperform the baselines on hard reward setting. Table 5 shows the performance at different percentiles (50% percentile being the median), where the BT variants (full and partial initialisation) outperform their non-BT counterparts in all lower percentiles (40, 20, 10 and 5%). Furthermore, BT variants show superior performance when considering reward functions not well aligned with the pre-trained policy (a set that is not well represented in Atari-57).
>
> In short, we believe that the proposed method is more general as it finds a better compromise than fine-tuning over the Atari-57 suite, achieving a strong performance across a wider range of settings.
>
> We will re-write the sub-section “Combining pre-trained behavior and weights” of Section 5 to clarify these important points.
>
> > The direct impact of this work is also severely limited by the execution and scope of its experiments: aside from not controlling for the use of intrinsic motivation/exploration objectives, it's not clear whether the effective use of pre-trained policies for exploration is specific to pre-training objectives that focus on encouraging exploration.
>
> We addressed the lack of comparisons to methods using intrinsic motivation from scratch in our answer above.
>
> We believe that the pre-training objective is crucial for obtaining good results with BT. We point out that in order to have “transferable behavior”, the pretraining objective needs to satisfy two general requirements: *(i)* it should scale gracefully with increased compute and data *(ii)* it should lead to a policy that produces complex behavior that may be leveraged in a subsequent transfer stage.
>
> NGU satisfies these requirements. The NGU intrinsic reward, was designed to scale and aims at producing policies that maximize the coverage of the environment. One contribution of our work is to consider the NGU scheme in the transfer learning setting.
>
> RND too was shown to scale to large distributed settings. While theoretically the intrinsic reward would vanish after pre-training, it is still able to produce policies showing interesting behavior in the unsupervised setting, [10] (reference from the manuscript). Thus, we incorporate it as a second alternative, to further show that the advantages of BT are not specific to use with NGU, but rather, with pre-training schemes that satisfy the desired criteria. It is out of the scope of this work to present a study on the benefits of different alternatives.
>
> We’d like to know if the reviewer considers that further experiments could help clarify this point.

---

> > ### Comment · Reviewer_FMRN · 2021-08-16
> > **Response to authors**
> >
> > The authors have comprehensively addressed my major concerns. I am updating my score from a 5 to a 6.
> >
> > Regarding the NGU baseline experiments, I presume that the frame count refers to supervised RL. It would be good to mention how many frames the BT run you compare to use during unsupervised pre-training.
> >
> > Regarding the interpretation of the metrics for assessing the performance of BT in comparison to alternatives, I strongly encourage the authors to 're-write the sub-section “Combining pre-trained behavior and weights” of Section 5 to clarify these important points'. Otherwise, I imagine I would hardly be the only reader to miss the nuance contained in results relegated to the supplementary material.

---

### Decision · Program_Chairs · 2021-09-27

**Decision:**

Reject

**Comment:**

This paper describes a simple but empirically effective strategy for unsupervised pre-training policies and utilizing them for exploration, with a temporally extended approach particularly effective for hard exploration challenges. The reviewers agree that the paper's concepts, while simple, are relevant and have potential to impact the community. Several reviewers raised concerns regarding the placement of the paper with respect to existing work and choices of experimental comparison. During responses, the authors presented additional detail (empirical and motivational) that has satisfied some, but not all reviewers.

In my opinion, this is a paper is just below the NeurIPS borderline, as the papers brings great performance on important problems, but does borrow elements heavily from existing approaches and gives less theory/analytical justification for its advances. Reviewers and authors have had some consensus about the value in further exploration of the multi-task properties, with potentially non-Atari setups or by finding more ways to run multi-task trials in modified Atari levels. I also encourage such exploration to more firmly establish the impact of the ideas.

A side note that has not continbuted to the decision: the term "transfer" led to some misunderstanding during our discussions. For some reviewers (and myself), transfer can refer to many sub-areas of RL where testing is not done in Atari: changes in robotic dynamics, sim2real, multi-task and more. Here "learning from policies learned in an unsupervised pre-trained fashion" would provide the needed context with certainty, despite being a mouthful.